# Modularity aided consistent attributed graph clustering via coarsening

**Samarth Bhatia\***       *samarth.bhatia23@alumni.iitd.ac.in*
*Indian Institute of Technology, Delhi*

**Yukti Makhija\***       *yukti.makhija@alumni.iitd.ac.in*
*Indian Institute of Technology, Delhi*

**Manoj Kumar**       *manoj.kumar@lnmiit.ac.in*
*LNM Institute of Technology, Jaipur*

**Sandeep Kumar**       *ksandeep@ee.iitd.ac.in*
*Indian Institute of Technology, Delhi*

*\* denotes equal contribution*

**Reviewed on OpenReview:** *https://openreview.net/forum?id=VtSIjrpFwA*

## Abstract

Graph clustering is an unsupervised learning technique for partitioning graphs with attributes and detecting communities. However, current methods struggle to accurately capture true community structures and intra-cluster relations, be computationally efficient, and identify smaller communities. We address these challenges by integrating coarsening and modularity maximization, effectively leveraging both adjacency and node features to enhance clustering accuracy. We propose a loss function incorporating log-determinant, smoothness, and modularity components using a block majorization-minimization technique, resulting in superior clustering outcomes. The method is theoretically consistent under the Degree-Corrected Stochastic Block Model (DC-SBM), ensuring asymptotic error-free performance and complete label recovery. Our provably convergent and time-efficient algorithm seamlessly integrates with Graph Neural Networks (GNNs) and Variational Graph AutoEncoders (VGAEs) to learn enhanced node features and deliver exceptional clustering performance. Extensive experiments on benchmark datasets demonstrate its superiority over existing state-of-the-art methods for both attributed and non-attributed graphs.

## 1 Introduction

Clustering is an unsupervised learning method that groups nodes together based on their attributes or graph structure, without considering node labels. This versatile approach has numerous applications in diverse fields, such as social network analysis (Tang et al., 2008), genetics, and bio-medicine (Cheng & Ma, 2022; Buterez et al., 2021), knowledge graphs (Hamaguchi et al., 2017), and computer vision (Mondal et al., 2021; Caron et al., 2018). The wide range of applications has led to the development of numerous graph clustering algorithms designed for specific challenges within these domains. State-of-the-art graph clustering methods predominantly fall into cut-based, similarity-driven, or modularity-based categories.

Cut-based methods (Wei & Cheng, 1989; Shi & Malik, 2000; Bianchi et al., 2020), aiming to minimize the number of edges (or similar metric) in a cut, may fall short in capturing the true community structure if the cut's edge count doesn't significantly deviate from random graph expectations. This approach originated from the Fiedler vector, which yields a graph cut with a minimal number of edges over all possible cuts (Fiedler, 1973; Newman, 2006a). Similarity-based techniques, reliant on pairwise node similarities, group nodes with

shared characteristics. These can be computationally intensive and susceptible to noise, yielding suboptimal results, especially in sparse or noisy data scenarios. Modularity-based methods rely on a statistical approach, measuring the disparity in edge density between a graph and a random graph with the same degree sequence. These modularity maximization methods exhibit a resolution limit (Fortunato & Barthélemy, 2007), as they may inadvertently lead to the neglect of smaller community structures within the graph. Each clustering approach, whether cut-based, similarity-based, or modularity-based, bears its own set of limitations that necessitates careful consideration based on the characteristics of the underlying graph data.

Moreover, graph reduction techniques such as coarsening, summarization, or condensation can also be utilized to facilitate the clustering task (Dhillon et al., 2007; Kumar et al., 2023; Loukas & Vandergheynst, 2018; Loukas, 2019). In the context of graph coarsening, the objective is to learn a reduced graph by merging similar nodes into supernodes. While this coarsening process is traditionally employed for graph reduction, it can be extended to clustering by reducing the original graph such that each class corresponds to a supernode. However, relying solely on coarsening may lead to suboptimal performance, particularly when the order of the reduction from the original graph size to the number of classes is large and results in a significant loss of information.

In this study, we introduce an optimization-based framework designed to enhance clustering by leveraging both the adjacency and the node features of the graph. Our proposed framework strategically incorporates coarsening and modularity maximization, refining partitioning outcomes and bolstering the effectiveness of the clustering process. We refer to our method as Q-MAGC, where Q denotes the use of modularity in the Modularity-Aided Graph Coarsening process. The algorithm minimizes a nuanced loss function, Q-MAGC objective, encompassing a log det term, smoothness, and modularity components, to ensure efficient clustering. Formulated as a multi-block non-convex optimization problem, our approach is adeptly addressed through a block majorization-minimization technique Razaviyayn et al. (2013), wherein variables are updated individually while keeping others fixed. The resulting algorithm demonstrates convergence, showcasing its efficacy in efficiently solving the proposed optimization problem.

To enhance the clustering performance, we embed our Q-MAGC objective function into various Graph Neural Network (GNN) architectures, introducing the Q-GCN algorithm. This novel approach elevates the quality of learned representations by leveraging the message passing and aggregation mechanisms of Graph Convolutional Networks (GCNs), ultimately improving the clustering outcomes. An additional feature of our technique is its capacity to explore inter-cluster relationships. The node attributes derived from the coarsened graph at the conclusion of the process serve as cluster embeddings, shedding light on the distinct characteristics of each cluster. Simultaneously, the edges within the coarsened graph unveil valuable insights into the relationships and connections between different clusters, providing a comprehensive understanding of the overall graph structure. This is a contributing factor to the improvement observed over existing methods and is particularly significant for conducting first-hand analyses on large unlabelled datasets.

We introduce two additional algorithms, Q-VGAE and Q-GMM-VGAE, incorporating variational graph auto-encoders to further enhance clustering accuracy. Through comprehensive experiments, we demonstrate the effectiveness of our proposed algorithms, surpassing the performance of state-of-the-art methods on synthetic and real-world benchmark datasets. Our approach showcases notable improvements in clustering performance, solidifying the robustness and superiority of our proposed methods.

**Key Contributions.**

- We present the first optimization-based framework for attributed graph clustering through coarsening via modularity maximization. Our approach demonstrates efficiency, theoretical convergence, and addresses limitations of existing methods. The paper offers comprehensive analysis and provides theoretical guarantees including KKT optimality, and convergence analysis which are often absent in prior research.

- We show that our method is theoretically (weakly and strongly) consistent under a Degree-Corrected SBM (DC-SBM) and shows asymptotically no errors (weakly consistent) and complete recovery of the original labels (strongly consistent). Proving consistency increases the reliability of our method and makes it more robust.

- We demonstrate the seamless integration of the proposed clustering objective with GNN-based architectures, leveraging message-passing to enhance our method, which is also backed up by experiments. [1]

- We perform thorough experimental validation on a diverse range of real-world and synthetic datasets, encompassing both attributed and non-attributed graphs of varying sizes. The results demonstrate the superior performance of our method compared to existing state-of-the-art approaches. We want to highlight that our method does not specialise for very large graphs, yet we present preliminary results in this regard.

- We conduct ablation studies to evaluate the behavior of the loss terms, compare runtime and complexities, and perform a comprehensive evaluation of modularity.

**Graph clustering setup.** Let $G = \{V, E, A, X\}$ be a graph with node set $V = \{v_1, v_2, ..., v_p\}$ ($|V| = p$), edge set $E \subset V \times V\}$($|E| = e$), weight (adjacency) matrix $A$ and node feature matrix $X \in \mathbb{R}^{p \times n}$. Also, let $\mathbf{d} = A \cdot \mathbb{1}_p \in \mathbb{Z}_+^p$ be the degree vector, where $\mathbb{1}_p$ is the vector of size $p$ having all entries 1. Then, the graph Laplacian is $\Theta = \mathrm{diag}(\mathbf{d}) - A$ and the set of all valid Laplacian matrices is defined as: $S_\Theta = \{\Theta \in \mathbb{R}^{p \times p} | \Theta_{ij} = \Theta{ji} \leq 0$ for $i \neq j, \Theta_{ii} = \sum_{j=1}^p \Theta_{ij}\}$. The goal of clustering is to learn a cluster assignment matrix $C \in \mathbb{R}_+^{p \times k}$, where $p$ is the number of nodes in the original graph and $k$ is the number of clusters. In the following section, we explore the concept of graph coarsening and examine its inherent connection to graph clustering.

## 2 Background and Related Works

In this section, we will discuss existing graph clustering techniques, how graph coarsening can be utilized for the graph clustering task and its limitations, and finally, the proposed formulation for graph clustering.

Graph clustering is a fundamental task in unsupervised learning, aiming to partition nodes into meaningful communities based on structural and attribute similarities. Graph clustering methods can be broadly classified into traditional and deep learning-based approaches. Traditional methods include cut-based, similarity-based, and modularity-based techniques. Cut-based methods (Wei & Cheng, 1989; Shi & Malik, 2000; Bianchi et al., 2020) minimize edge cuts to separate clusters but often fail to capture true communities. Similarity-based approaches group nodes based on pairwise similarity but suffer from high computational costs and noise sensitivity. Modularity-based methods (Guimerà & Nunes Amaral, 2005; Salha-Galvan et al., 2022; Bhowmick et al., 2023; Tsitsulin et al., 2023) optimize modularity to enhance community detection but face resolution limits that hinder the identification of smaller clusters (Fortunato & Barthélemy, 2007).

Next, deep learning-based methods leverage neural networks for graph clustering. Graph Autoencoders (GAEs) and Variational Graph Autoencoders (VGAEs) (Kipf & Welling, 2016a) learn latent node representations through adjacency reconstruction. Contrastive learning methods (Hassani & Khasahmadi, 2020; Zhao et al., 2021; Liu et al., 2023a) improve these representations by distinguishing similar and dissimilar node pairs. GMM-VGAE (Hui et al., 2020) refines clustering by incorporating a Gaussian Mixture Model to model complex data distributions. Despite their effectiveness, deep learning-based methods struggle with scalability, high computational complexity, and instability in graph clustering. Furthermore, graph coarsening is also utilized to perform the clustering task. In the next subsection, we will discuss how graph coarsening can be leveraged for graph clustering. A more in-depth literature survey on such methods can be found in Appendix B.

### 2.1 Graph Coarsening

Graph coarsening is a graph dimensionality reduction technique used in large-scale machine learning to construct a smaller or coarsened graph $G_c$ from the original graph $G = \{V, E, A, X\}$ while preserving properties of the original graph $G$. Graph coarsening aims to learn a mapping matrix $C \in \mathbb{R}_+^{p \times k}$, where $p$ is the number of nodes in the original graph and $k$ is the number of nodes in the coarsened graph. Each non-zero entry of the mapping matrix $C$, i.e., $C_{ij}$, indicates that the $i$-th node of $G$ is mapped to the $j$-th supernode. For $C \in \mathcal{C}$, the relationship between the original graph Laplacian $\Theta$, the coarsened graph Laplacian $\Theta_c$, and the mapping matrix $C$ is given by $\Theta_c = C^T \Theta C$. Also, the coarsened graph feature matrix is computed using

---

[1]Refer to Appendix A for the code

the relation $X_c = C^\dagger X$. Moreover, for a balanced mapping, the mapping matrix $C$ belongs to the following set (Kumar et al., 2023; Loukas & Vandergheynst, 2018; Loukas, 2019):

$$\mathcal{C} = \left\{ C \in \mathbb{R}_+^{p \times k} | \ \langle C_i, C_j \rangle = 0 \ \forall \ i \neq j, \quad \langle C_i, C_i \rangle = d_i, \|C_i\|_0 \geq 1 \text{ and } \left\| [C^\top]_i \right\|_0 = 1 \right\} \tag{1}$$

Graph coarsening can be extended to graph clustering by reducing the size of the coarsened graph to the number of classes ($k$). However, in most graphs, the number of classes is very small, and reducing the graph to this extent may lead to poor clustering quality. Before moving towards the problem formulation, we will also discuss spectral modularity.

## 2.2 Spectral Modularity Maximization

Spectral Clustering is the most direct approach to graph clustering, where we minimize the volume of inter-cluster edges. Modularity, introduced in Newman (2006b), is the difference between the number of edges within a cluster $\mathbf{C_i}$ and the expected number of such edges in a random graph with an identical degree sequence. It is mathematically defined as:

$$\mathcal{Q} = \frac{1}{2e} \sum_{i,j=1}^{k} \left[ A_{ij} - \frac{d_i d_j}{2e} \right] \delta(c_i, c_j) \tag{2}$$

where $\delta(c_i, c_j)$ is the Kronecker delta between clusters $i$ and $j$. Maximizing this form of modularity is NP-hard (Brandes et al., 2008). However, we can approximate it using a spectral relaxation, which involves a modularity matrix $B$. The modularity matrix $B$ and spectral modularity are defined as follows:

$$B = A - \frac{\mathbf{d}\mathbf{d}^T}{2e}, \qquad \mathbf{d} = A \cdot \mathbb{1}_p, \qquad \mathcal{Q} = \frac{1}{2e} Tr(C^T B C) \tag{3}$$

$B$ is symmetric and is defined such that its row-sums and column-sums are zero, thereby making $\mathbf{1}$ one of its eigenvectors and 0 the corresponding eigenvalue. These spectral properties of the modularity matrix are also observed in the Laplacian, as noted in Newman (2006b), which is a crucial element in spectral clustering. Modularity is maximized when $u_1^T s$ is maximized, where $u$ are the eigenvectors of $B$ and $s$ is the community assignment vector, i.e., placing the majority of the summation in $Q$ on the first (and largest) eigenvalue of $B$. Moreover, modularity is closely associated with community detection. These special spectral properties make $B$ an ideal choice for graph clustering.

Next, we will discuss existing graph clustering studies that apply graph coarsening techniques or utilize modularity maximization. MinCutPool (Bianchi et al., 2020) formulates a differentiable relaxation of spectral clustering via pooling by jointly optimizing over the clustering objective with an orthogonal regularization. DiffPool (Ying et al., 2018) learns soft cluster assignments at each layer of the GNN and optimizes two additional losses, an entropy to penalize the soft assignments and a link prediction based loss. Next, SAGPool (Lee et al., 2019) calculates attention scores and node embeddings to determine the nodes that need to be preserved or removed. Some disadvantages of these methods are instability and computational complexity in the case of SAGPool and DiffPool and convergence in MinCutPool. Various heuristic algorithms have been established that solve the NP-hard problem of modularity maximization including sampling, simulated annealing (Guimerà & Nunes Amaral, 2005; Guimerà & Amaral, 2005) and the Newman-Girvan algorithm with $\mathcal{O}(p^3)$ time complexity. Greedy algorithms (Louvain/Leiden) (Newman, 2004; Blondel et al., 2008) improve on this. These algorithms require intensive compute and don't use node features. Modularity maximization using GNNs has also garnered attention recently. DMoN (Tsitsulin et al., 2023) optimizes only for modularity with a collapse regularization to prevent the trivial solution, but offers no theoretical guarantees about convergence. A more comprehensive literature review can be found in Appendix B.

Current graph clustering methods often struggle to accurately capture both intra-cluster and inter-cluster relationships, achieve computational efficiency, and identify smaller communities, thereby limiting their effectiveness. These methods face significant challenges including instability and the lack of convergence guarantees, especially when employing coarsening techniques. Modularity maximization, despite extensive study, relies on computationally intensive heuristics and lacks theoretical convergence guarantees.

### 2.3 Proposed Problem Formulation

By incorporating both adjacency and node features, our approach aims to robustly capture intra-cluster and inter-cluster dynamics. This framework ensures stability, guarantees convergence, and consistently delivers superior clustering results with enhanced computational efficiency compared to existing methods. Given original graph $G(V, E, A, X)$, the proposed optimization formulation is:

$$\min_C \mathcal{L}_{MAGC} = f(C, X, \Theta) + g(C, A) + h(C, \Theta)$$

$$\text{subject to } C \in \mathcal{S}_C = \{C \in \mathbb{R}^{p \times k} | C \geq 0, \ \left\| [C^T]_i \right\|_2^2 \leq 1\} \ \forall \ i = 1, 2 \ldots p \tag{4}$$

Here, $C \in \mathbb{R}_+^{p \times k}$ represents the clustering matrix to be learned, where each non-zero entry $C_{ij}$ indicates that the $i$-th node is assigned to the $j$-th cluster. The term $f(C, X, \Theta)$ denotes the graph coarsening objective which compresses similar nodes together while preserving structural properties. The function $g(C, A)$ represents the modularity objective, improving clustering performance. Additionally, $h(C, \Theta)$ is a regularization term enforcing desirable properties in the clustering matrix, as defined in Equation 1. The overarching goal of the optimization problem 4 is to choose $f(C, X, \Theta)$, $g(C, A)$, and $h(C, \Theta)$ such that nodes are optimally clustered and inter-cluster connectivity is maintained. In the next section, we will develop the clustering algorithm, leveraging the coarsening objective and modularity to enhance clustering performance.

## 3 MAGC Algorithm

Given a graph $G = \{V, E, A, X\}$, considering $f(C, X_C, \Theta) = \text{tr}(X_C^T C^T \Theta C X_C)$, $g(C, A) = -\frac{\beta}{2e} \text{tr}(C^T B C)$ (where $B = A - \frac{\mathbf{d}\mathbf{d}^T}{2e}$, $\mathbf{d} = A \cdot \mathbb{1}$), and $h(C, \Theta) = -\gamma \log \det(C^T \Theta C + J)$, to obtain clustering matrix $C$ we formulate the following optimization problem :

$$\min_{X_C, C} \mathcal{L}_{MAGC} = \text{tr}(X_C^T C^T \Theta C X_C) - \frac{\beta}{2e} \text{tr}(C^T B C) - \gamma \log \det(C^T \Theta C + J)$$

$$\text{subject to } C \in \mathcal{S}_C = \{C \in \mathbb{R}^{p \times k} | \ \left\| C_i^T \right\|_2^2 \leq 1\} \forall i, X = C X_C \ \text{where, } J = \frac{1}{k} \mathbf{1}_{k \times k} \tag{5}$$

The term $\text{tr}(X_C^T C^T \Theta C X_C)$ represents the smoothness or Dirichlet energy of the coarsened graph while $C^T \Theta C$ is the Laplacian matrix of the coarsened graph. Minimizing smoothness ensures that the clusters or supernodes with similar features are linked with stronger weights. The term $\text{tr}(C^T B C)$ corresponds to the graph's modularity, enhancing the quality of the clusters formed. The term $-\log \det(C^T \Theta C + J)$ is crucial for maintaining inter-cluster edges in the coarsened graph. For a connected graph matrix with $k$ super-nodes or clusters, the rank of $C^T \Theta C$ is $k - 1$. Adding $J$ to $C^T \Theta C$ makes $C^T \Theta C + J$ a full-rank matrix without altering the row and column space of $C^T \Theta C$ (Kumar et al., 2020). This is ensured because it can be written as $-\sum_i \log \lambda_i$ where $\lambda_i$'s are the eigenvalues of the $\Theta_C$ - thus, minimizing this would result in minimizing the multiplicity of 0-eigenvalues, in turn minimizing the number of connected components (which can't be less than 1).

Problem 5 is a multi-block non-convex optimization problem solved using the Block Successive Upper-bound Minimization (BSUM) framework. All terms except modularity are convex in nature, which we prove in Appendix C. We iteratively update $C$ and $X_C$ alternately while keeping the other constant. This process continues until convergence or the stopping criteria are met. Since the constraint $X = C X_C$ is hard and difficult to enforce, we relax it by adding the term $\frac{\alpha}{2} \|X - C X_C\|_F^2$ to the objective. This term ensures that each node is assigned to a cluster, leaving no node unassigned. Additionally, when needed, we can add an optional sparsity regularization term $\lambda \|C^T\|_{1,2}^2$, which can be seen as ensuring that each node is assigned to exactly one cluster, avoiding any overlap in node assignments across clusters.

**Updating $C$**

Treating $X_C$ as constant and $C$ as a variable the sub-problem for $C$ is:

$$\min_{C} f(C) = \operatorname{tr}(X_C^T C^T \Theta C X_C) - \frac{\beta}{2e}\operatorname{tr}(C^T BC) - \gamma \log \det(C^T \Theta C + J) + \frac{\alpha}{2}\|X - CX_C\|_F^2$$

$$\text{subject to } C \in \mathcal{S}_C = \{C \in \mathbb{R}^{p \times k}|C \geq 0, \ \|C_i^T\|_2^2 \leq 1\} \ \forall i, \text{ where, } J = \frac{1}{k}\mathbf{1}_{k \times k} \tag{6}$$

By using the first-order Taylor series approximation, a majorised function for $f(C)$ at $C^t$ ($C$ after $t$ iterations) can be obtained as:

$$g(C|C^t) = f(C^t) + \nabla f(C^t) \cdot (C - C^t) + \frac{L}{2}\|C - C^t\|^2 \tag{7}$$

where $f(C)$ is $L-$Lipschitz continuous gradient function $L = \max(L_1, L_2, L_3, L_4)$ with $L_1, L_2, L_3, L_4$ the Lipschitz constants of $-\gamma \log \det(C^T \Theta C + J)$, $\operatorname{tr}(X_C^T C^T \Theta C X_C)$, $\|CX_C - X\|_F^2$, $\operatorname{tr}(C^T BC)$, respectively. We prove this in Appendix D. We can expand this as:

$$g(C|C^t) = f(C^t) + \nabla f(C^t) \cdot (C - C^t) + \frac{L}{2}\operatorname{tr}\big((C - C^t)^T(C - C^t)\big) \tag{8}$$

$$= f(C^t) + \nabla f(C^t) \cdot (C - C^t) + \frac{L}{2}\operatorname{tr}(C^T C) - L\operatorname{tr}(C^T C^t) + \frac{L}{2}\operatorname{tr}((C^t)^T C^t)$$

Ignoring constant terms we get

$$= \operatorname{tr}(C^T \nabla f(C^t)) - L\operatorname{tr}(C^T C^t) + \frac{L}{2}\operatorname{tr}(C^T C) \tag{9}$$

Now, the majorised problem of Equation 7 becomes

$$\min_{C \in \mathcal{S}_C} \operatorname{tr}(\frac{1}{2}C^T C - C^T\Big(C^t - \frac{1}{L}\nabla f(C^t)\Big)) \tag{10}$$

The optimal solution to Equation 10, found by using Karush–Kuhn–Tucker (KKT) optimality conditions is (Proof is deferred to the Appendix D):

$$C^{t+1} = \Big(C^t - \frac{1}{L}\nabla f\big(C^t\big)\Big)^+ \tag{11}$$

$$\text{where, } \nabla f\big(C^t\big) = -2\gamma \Theta C^t(C^{t^T}\Theta C^t + J)^{-1} + \alpha(C^t X_C - X)X_C^T + 2\Theta C^t X_C X_C^T - \frac{\beta}{e}BC^t \tag{12}$$

**Updating $X_C$**

Treating $C$ fixed and $X_c$ as variable. The subproblem for updating $X_c$ is

$$\min_{\tilde{X}} f(\tilde{X}) = \operatorname{tr}(X_C^T C^T \Theta C X_C) + \frac{\alpha}{2}\|X - CX_C\|_F^2 \tag{13}$$

The closed form solution of problem Equation 13 can be obtained by putting the gradient of $f(\tilde{X})$ to zero.

$$X_C^{t+1} = \Big(\frac{2}{\alpha}C^T \Theta C + C^T C\Big)^{-1}C^T X \tag{14}$$

**Convergence Analysis**

**Theorem 1.** *The sequence $\{C^{t+1}, X_C^{t+1}\}$ generated by Algorithm 1 converges to the set of Karush–Kuhn–Tucker (KKT) optimality points for Problem 5*

*Proof Sketch.* We derive the majorized problem for L-Lipschitz continuous and differentiable functions. Then, we formulate the Lagrangian (Equation 25) of the majorized problem Equation 10 at the $t^{th}$ iteration and define the dual variables. The next step involves solving the KKT conditions that satisfy primal and dual feasibility, complementary slackness, and setting the gradient to zero. This allows us to obtain the optimal solution for $C^{(t+1)}$. The detailed proof can be found in the Appendix E. □

**Complexity Analysis.** The worst-case time complexity of a loop (i.e. one epoch) in Algorithm 1 is $\mathcal{O}(p^2 k + pkn)$ because of the matrix multiplication in the update rule of $C$ Equation 11. Here, $k$ is the number of clusters and $n$ is the feature dimension. Note that $k$ is much smaller than both $p$ and $n$. This makes our method much faster than previous optimization based methods and faster than GCN-based clustering methods which have complexities around $\mathcal{O}(p^2 n + pn^2)$. We discuss this more in Appendix F.

---

**Algorithm 1** Q-MAGC Algorithm

---

**Require:** $G(X, \Theta), \alpha, \beta, \gamma, \lambda$

1: $t \leftarrow 0$
2: **while** Stopping Criteria not met **do**
3:      Update $C^{t+1}$ as in Equation 11
4:      Update $X_C^{t+1}$ as in Equation 14
5:      $t \leftarrow t + 1$
6: **end while**
7: **return** $C^t, X_C^t$

---

**Consistency Analysis on Degree Corrected Stochastic Block Models (DC-SBM).**

To evaluate the robustness of the proposed optimization objective, we aim to prove its consistency. By consistency we mean the objective can recover the cluster assignments correctly. Strong consistency refers to perfect recovery of cluster assignments, while weak consistency allows for $\epsilon$-margin of error in the assignments. To check whether the proposed objective (Equation 5) results in consistent clustering, we need to assume a random graph model. In this section, we prove our method is weakly and strongly consistent under Degree Corrected Stochastic Block Models (DC-SBM).

A graph $G(V, E)$ is generated by a DC-SBM with $p$ nodes which belong to $k$ classes. Following the setup in Zhao et al. (2012), each node $v_i$ is associated with a label-degree pair $(y_i, t_i)$ drawn from a discrete joint distribution $\Pi_{k \times m}$. Here, $t_i$ takes values $0 \leq x_1 \leq \cdots \leq x_m$ with $\mathbb{E}[t_i] = 1$, and $y_i \in [k]$. Additionally, we have a symmetric $k \times k$ matrix $P$ that specifies the probabilities of inter-cluster edges. The edges between each node pair $(v_i, v_j)$ are sampled as independent Bernoulli trials with probability $\mathbb{E}[A_{ij}] = t_i t_j P_{y_i y_j}$. To ensure $\mathbb{E}[A_{ij}] < 1$, DC-SBMs must satisfy $x_m^2 (\max_{i,j} P_{ij}) \leq 1$. The matrix $P$ is allowed to scale with the number of nodes $p$ and is reparameterized as $P_p = \rho_p P$, where $\rho_p = \Pr[A_{ij} = 1] \to 0$ as $p$ increases, and $P$ is fixed. The average degree of the DC-SBM graph is defined as $\lambda_p = p \rho_p$.

We select DC-SBM for our analysis because it incorporates a degree parameter for each node, which accommodates the heterogeneity of real-world graphs. This feature allows for high-degree nodes at the cluster centers, a common characteristic in practical graph structures. While DC-SBM effectively models real-world graphs, it remains analytically tractable, making it a good choice for evaluating clustering behaviors.

Let $\hat{y}_i$ denote the predicted cluster for node $v_i$. The assignment matrix $C$ is the one-hot encoding of $y$, such that $C_i = \text{one-hot}(y_i)$. We define $O(e) \in \mathbb{Z}_{\geq 0}^{k \times k}$ as the inter-cluster edge count matrix for a given cluster assignment $e \in [k]^p$. Here, $O_{ql}(e) = \sum_{ij} A_{ij} \mathbb{1}\{e_i = q, e_j = l\}$ denotes the number of edges between clusters $q$ and $l$. Also, We represent the class frequency distribution as $\pi \in [0, 1]^k$, where $\pi_i$ indicates the fraction of nodes assigned to cluster $i$.

**Definition 1.** *(Strong and Weak Consistency). The clustering objective is defined to be strongly consistent if* $\lim_{p \to \infty} Pr[\hat{y} = y] \to 1$. *A weaker notion of consistency is defined by* $\lim_{p \to \infty} Pr\left[\frac{1}{p} \sum_{i=1}^{p} \mathbb{1}\{\hat{y} \neq y\} < \epsilon\right] \to 1 \ \ \forall \ \epsilon > 0$.

Zhao et al. (2012) prove the consistency of multiple clustering objectives under the DC-SBM, including Newman–Girvan modularity (Theorem 3.1). They also provide a general theorem (Theorem 4.1) for checking consistency under DC-SBMs for any criterion $\mathcal{L}(e)$, which can be expressed as $\mathcal{L}(e) = F\left(\frac{O(e)}{\mu_p}, \pi\right)$, where $\mu_p = p^2 \rho_p$ and $e \in [k]^p$ is a cluster assignment.

The consistency of an objective is evaluated by determining if the *population version* of $\mathcal{L}(e)$ is maximized by the true cluster assignment $y$. The *population version* of $\mathcal{L}(e)$ is obtained by taking its conditional expectations given $y$ and $t$. We consider an array $S \in \mathbb{R}^{k \times k \times m}$ and define a matrix $H(S) \in \mathbb{R}^{k \times k}$ as $H_{kl}(S) = \sum_{abuv} x_u x_v P_{ab} S_{kau} S_{lbv}$. Additionally, we define a vector $h(S) \in \mathbb{R}^k$ as $h_k(S) = \sum_{au} S_{kau}$. Here, $H(S)$ and $h(S)$ denote the *population versions* of $O(e)$ and $\pi$, respectively, such that $\frac{1}{\mu_p} \mathbb{E}[O|C, t] = H(S)$ and $\mathbb{E}[\pi|C, t] = h(S)$.

**Lemma 1.** *(Theorem 4.1 from Zhao et al. (2012)) For any $\mathcal{L}(e) = F(\frac{O(e)}{\mu_p}, f(e))$, if $F$ is uniquely maximized by $S = \mathbb{D}$ and $\pi, P, F$ satisfy the regularity conditions, then $\mathcal{L}$ is strongly consistent under DC-SBMs if $\frac{\lambda_p}{\log p} \to \infty$ and weakly consistent if $\lambda_p \to \infty$:*

1. *$F$ is Lipschitz in its arguments $(H(S), h(S))$*
2. *The directional derivative $\frac{\partial^2}{\partial \varepsilon^2} F(M_0 + \varepsilon(M_1 - M_0), \mathbf{t_0} + \varepsilon(\mathbf{t_1} - \mathbf{t_0}))\big|_{\varepsilon=0^+}$ is continuous in $(M_1, \mathbf{t_1})$ for all $(M_0, \mathbf{t_0})$ in a neighborhood of $(H(\mathbb{D}), \pi)$*
3. *With $G(S) = F(H(S), h(S))$,      $\frac{\partial G((1-\varepsilon)\mathbb{D} + \varepsilon S)}{\partial \varepsilon}|_{\varepsilon=0^+} < -C < 0 \ \forall \ \pi, P$*

It is important to note that the paper focuses on maximizing an objective function. Therefore, we consider the negative of our loss function, $-\mathcal{L}_{MAGC}$. Additionally, we only consider the objective function defined in Equation 5, as regularizers can be adjusted for different downstream tasks.

**Theorem 2.** *Under the DC-SBM, $\mathcal{L}_{MAGC}$ is strongly consistent when $\lambda_p/\log p \to \infty$ and weakly consistent when $\lambda_p \to \infty$.*

*Proof Sketch.* The proof is divided into two parts. First, we demonstrate that $\mathcal{L}_{MAGC}$ can be expressed in terms of $O$ and $\pi$(the class frequency). We do this by establishing relations between the various terms made up of $C$, $A$, $D$ and $\Theta$. We show that $O$ is equivalent to the the cluster-level adjacency matrix. Combining these relations, we are able to express $\mathcal{L}_{MAGC}$ in the required form. Applying Lemma 1, we show that $\mathcal{L}_{\text{MAGC}}$ must be uniquely minimized at any point $(y^*, A^*)$ s.t. $\mathbb{E}_p[\pi(y_p)] = \pi(y^*)$ and $\mathbb{E}_p[A^{(p)}] = A^*$. Second, we verify that, in this form, the *population version* of $\mathcal{L}_{MAGC}$ (Equation 51) satisfies the regularity conditions specified in Lemma 1. It is important to note that DC-SBMs do not consider node features, so we treat $X$ and $X_C$ as constants. We defer the detailed proof to Appendix G. $\qquad\square$

We also demonstrate experimental consistency of the optimization objective Equation 5 in Appendix H, resulting in complete recovery of the labels. This consistency extends to the objective with additional regularization terms, as detailed therein. We want to emphasize that the consistency analysis for Equation 5 holds regardless of how the objective is optimized: whether by integrating our loss into Graph Neural Networks (GNNs) or by employing Block Majorization-Minimization techniques.

## 4 Integrating with GNNs

Our optimization framework integrates seamlessly with Graph Neural Networks (GNNs) by incorporating the objective Equation 5 into the loss function. This integration can be minimized using gradient descent. We demonstrate the effectiveness of this approach on several popular GNN architectures, including Graph Convolutional Networks (GCNs) (Kipf & Welling, 2016b), Variational Graph Auto-Encoders (VGAEs) (Kipf & Welling, 2016a), and a variant known as Gaussian Mixture Model VGAE (GMM-VGAE) (Hui et al., 2020).

---

**Algorithm 2** Q-GCN Algorithm

---

**Require:** $G(X, \Theta), \alpha, \beta, \gamma, \lambda$
**Require:** $k$ GCN layers - weights $W^{(1..k)}$
1: $t \leftarrow 0$
2: **while** Stopping Criteria not met **do**
3:      $C^{t+1} \leftarrow \text{GCN}(X, A)$
4:      $X_C^{t+1} \leftarrow C^\dagger X$
5:      $t \leftarrow t + 1$
6:      $\mathcal{L} \leftarrow \mathcal{L}_{\text{MAGC}}(C, X)$
7: **end while**
8: **return** $C^t, X_C^t$

---

We iteratively learn the matrix $C$ using gradient descent. From Equation 1, we update $X_C$ using the relation $X_C = C^\dagger X$. It is important to note that the loss term $CX_C - X$ will not necessarily be zero. This arises because we use a "soft" version of $C$ ($C_{i,j}$ is the probability $\in [0,1]$) in the loss function to enable gradient flow, while a "hard" version of $C$ ($C_{i,j}$ is the binary assignment $\in \{0,1\}$) is used in the update step. This method ensures that $C$ naturally becomes harder and exhibits a higher prediction probability.

**Q-GCN.** We integrate our loss function ($\mathcal{L}_{MAGC}$, as defined in Equation 5) into a simple three-layer Graph Convolutional Network (GCN) model. The soft cluster assignments $C$ are learned as the output of the final GCN layer. The detailed architecture and loss function are illustrated in Figure 1.

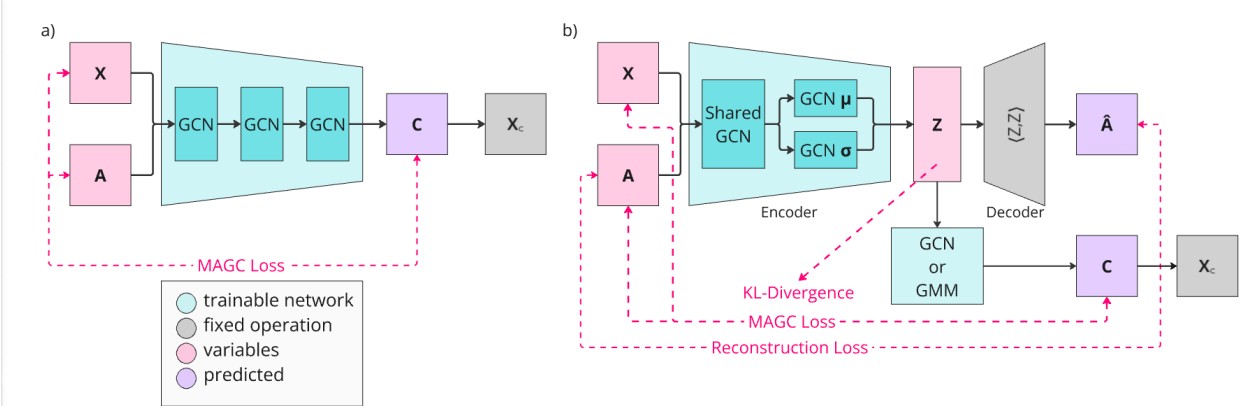

Figure 1: **a) Architecture of Q-GCN.** We want to train the encoder to learn the soft cluster assignment matrix $C$. The coarsened features $X_C$ are obtained using the relation $X_C^{t+1} = C^{t+1^\dagger} X$. Finally, our proposed MAGC loss is then computed using $C$ and $X_C$.
**b) Architecture of Q-VGAE/Q-GMM-VGAE.** The three-layer GCN encoder takes $X$ and $A$ as inputs to learn the latent representation $Z$ of the graph. $Z$ is then passed through an inner-product decoder to reconstruct the adjacency matrix $\hat{A}$. The reconstruction loss is calculated between $\hat{A}$ and $A$, and the KL-divergence is applied to $Z$. In Q-VGAE (or Q-GMM-VGAE), $Z$ is also passed through a GCN layer (or GMM) to output the soft cluster assignments $C$. The MAGC loss is then computed as in Q-GCN.

**Q-VGAE.** The VGAE loss can be written as

$$\mathcal{L}_{VGAE} = \lambda_{recon} \underbrace{\mathbb{E}_{q(Z|X,A)}[\log p(\hat{A}|Z)]}_{\text{Reconstruction Error}} - \lambda_{kl} \underbrace{\text{KL}[q(Z|X,A) \ || \ p(Z)]}_{\text{Kullback-Leibler divergence}}$$

where, $Z$ represents the latent space of the VGAE, $\hat{A}$ is the reconstructed adjacency matrix, and $\lambda_{recon}$ and $\lambda_{kl}$ are hyperparameters. We add a GCN layer on top of this architecture, which takes $Z$ as input and predicts $C$. A detailed summary of the VGAE loss terms is provided in Appendix I. For the VGAE, we minimize the sum of three losses: the reconstruction loss, the KL-divergence loss, and our loss. This combined loss function is expressed as:

$$\mathcal{L}_{Q-VGAE} = \mathcal{L}_{MAGC} + \mathcal{L}_{VGAE}$$

**Q-GMM-VGAE.** This variant of VGAE incorporates a Gaussian Mixture Model (GMM) in the latent space to better capture data distributions. This approach is effective because it minimizes the evidence lower bound (ELBO) or variational lower bound (Hui et al., 2020; Kingma & Welling, 2014; Kipf & Welling, 2016a) using multiple priors, rather than a single Gaussian prior as in standard VGAE. Hui et al. (2020) use a number of priors equal to the number of clusters.

---

**Algorithm 3** Q-VGAE/Q-GMM-VGAE Algorithm $Z$ is the latent space of the VGAE and $\hat{A}$ is the reconstructed adjacency matrix.

---

**Require:** $G(X, \Theta), \alpha, \beta, \gamma, \lambda$
**Require:** Variational Encoder - VarEnc (GCN, $\mu, \sigma$)
**Require:** Prediction Head - Pred (GCN or GMM)
1:  $t \leftarrow 0$
2:  **while** Stopping Criteria not met **do**
3:      $Z \leftarrow \text{VarEnc}_{\mu,\sigma}(X, A)$
4:      $\hat{A} \leftarrow ZZ^T$
5:      $C^{t+1} \leftarrow \text{Pred}(Z, A)$
6:      $X_C^{t+1} \leftarrow C^\dagger X$
7:      $t \leftarrow t + 1$
8:      $\mathcal{L} \leftarrow \mathcal{L}_{\text{MAGC}}(C, X) + \mathcal{L}_{\text{VGAE}}(X, A, Z, \hat{A})$
9:  **end while**
10: **return** $C^t, X_C^t$

## 5 Experiments

### 5.1 Benchmark Datasets and Baselines

We evaluate our method on a diverse set of datasets, including small attributed datasets like Cora and CiteSeer, larger datasets like PubMed, and unattributed datasets such as Airports (Brazil, Europe, and USA). A summary of these can be seen in Table 1. Additionally, we test our method on very large datasets like CoauthorCS/Physics, AmazonPhoto/PC, and ogbn-arxiv. A detailed summary of all the datasets used is provided in Appendix J.

To assess the performance of our method, we compare it against three types of state-of-the-art methods based on the input and architecture type: methods that use only node attributes, methods that use only graph structure, and methods that use both graph structure and node

| Name | p ($|\mathbf{V}|$) | n ($|X_i|$) | e ($|E|$) | k ($y$) |
|------|------|------|------|------|
| Cora | 2708 | 1433 | 5278 | 7 |
| CiteSeer | 3327 | 3703 | 4614 | 6 |
| PubMed | 19717 | 500 | 44325 | 3 |
| Brazil | 131 | 0 | 1074 | 4 |
| Europe | 399 | 0 | 5995 | 4 |
| USA | 1190 | 0 | 13599 | 4 |

Table 1: Datasets summary.

attributes. The last category is further subdivided into graph coarsening methods, GCN-based architectures, VGAE-based architectures and contrastive methods, and heavily modified VGAE architectures. This comprehensive evaluation allows us to demonstrate the robustness and versatility of our approach across various data and model configurations. The details on hyperparameter tuning can be found in Appendix subsection K.1.

### 5.2 Metrics

To evaluate the performance of our method, we utilize label alignment metrics that compare ground truth node labels with cluster assignments. Specifically, we measure Normalised Mutual Information (NMI), Adjusted Rand Index (ARI), and Accuracy (ACC), with higher values indicating superior performance. For detailed explanations of these metrics, please refer to Appendix J. We selected NMI as the primary metric for evaluating model performance based on its prominence in graph clustering literature and its comprehensive ability to assess the quality of our cluster assignments. Training Details are available in the Appendix L.

### 5.3 Attributed Graph Clustering

We present our key results on the real datasets Cora, CiteSeer, and PubMed in Table 2. Our proposed method outperforms all existing methods in terms of NMI and demonstrates competitive performance in Accuracy and ARI. The best models were selected based on NMI scores. Results for very large datasets are provided in Appendix M. Unlike some methods such as S3GC (Devvrit et al., 2022), which use randomly-sampled batches that can introduce bias by breaking community structure, we perform full-batch training by passing the entire graph. For extremely large graphs, such as ogbn-arxiv, we have also utilized batching.

### 5.4 Non-Attributed Graph Clustering

Our results in Table 3 demonstrate that our method achieves competitive or superior performance in terms of NMI, even for non-attributed datasets. For these, we use a one-hot encoding of the degree vector as features. While this is a basic approach compared to learning-based methods like DeepWalk (Perozzi et al., 2014b) and node2vec (Grover & Leskovec, 2016), it ensures a fair comparison since other methods also use this feature representation.

| Method | Cora | | | CiteSeer | | | PubMed | | |
|---|---|---|---|---|---|---|---|---|---|
| | ACC ↑ | NMI ↑ | ARI ↑ | ACC ↑ | NMI ↑ | ARI ↑ | ACC ↑ | NMI ↑ | ARI ↑ |
| K-means | 34.7 | 16.7 | 25.4 | 38.5 | 17.1 | 30.5 | 57.3 | 29.1 | 57.4 |
| Spectral Clustering | 34.2 | 19.5 | 30.2 | 25.9 | 11.8 | 29.5 | 39.7 | 3.5 | 52.0 |
| DeepWalk (Perozzi et al., 2014a) | 46.7 | 31.8 | 38.1 | 36.2 | 9.7 | 26.7 | 61.9 | 16.7 | 47.1 |
| Louvain (Blondel et al., 2008) | 52.4 | 42.7 | 24.0 | 49.9 | 24.7 | 9.2 | 30.4 | 20.0 | 10.3 |
| GAE [NeurIPS'16] (Kipf & Welling, 2016a) | 61.3 | 44.4 | 38.1 | 48.2 | 22.7 | 19.2 | 63.2 | 24.9 | 24.6 |
| DGI [ICLR'19] (Veličković et al., 2019) | 71.3 | 56.4 | 51.1 | 68.8 | 44.4 | 45.0 | 53.3 | 18.1 | 16.6 |
| GIC [PAKDD'21] (Mavromatis & Karypis, 2021) | 72.5 | 53.7 | 50.8 | 69.6 | 45.3 | 46.5 | 67.3 | 31.9 | 29.1 |
| DAEGC [IJCAI'19] (Wang et al., 2019) | 70.4 | 52.8 | 49.6 | 67.2 | 39.7 | 41.0 | 67.1 | 26.6 | 27.8 |
| GALA [ICCV'19] (Park et al., 2019) | **74.5** | 57.6 | 53.1 | 69.3 | 44.1 | 44.6 | 69.3 | 32.1 | 32.1 |
| AGE [KDD'20] (Cui et al., 2020) | 73.5 | 57.5 | 50.0 | 69.7 | 44.9 | 34.1 | **71.1** | 31.6 | **33.4** |
| DCRN [AAAI'22] (Liu et al., 2022) | 61.9 | 45.1 | 33.1 | 70.8 | 45.8 | 47.6 | 69.8 | 32.2 | 31.4 |
| FGC [JMLR'23] (Kumar et al., 2023) | 53.8 | 23.2 | 20.5 | 54.2 | 31.1 | 28.2 | 67.1 | 26.6 | 27.8 |
| **Q-MAGC (Ours)** | 65.8 | 51.8 | 42.0 | 65.9 | 40.8 | 40.1 | 66.7 | **32.8** | 27.9 |
| **Q-GCN (Ours)** | 71.6 | **58.3** | **53.6** | 71.5 | **47.0** | **49.1** | 64.1 | 32.1 | 26.5 |
| SCGC [IEEE TNNLS'23] (Liu et al., 2023a) | **73.8** | 56.1 | 51.7 | **71.0** | 45.2 | 46.2 | - | - | - |
| MVGRL [ICML'20] (Hassani & Khasahmadi, 2020) | 73.2 | 56.2 | 51.9 | 68.1 | 43.2 | 43.4 | 69.3 | **34.4** | **32.3** |
| VGAE [NeurIPS'16] (Kipf & Welling, 2016a) | 64.7 | 43.4 | 37.5 | 51.9 | 24.9 | 23.8 | **69.6** | 28.6 | 31.7 |
| ARGA [IJCAI'18] (Pan et al., 2018) | 64.0 | 35.2 | 61.9 | 57.3 | 34.1 | **54.6** | 59.1 | 23.2 | 29.1 |
| ARVGA [IJCAI'18] (Pan et al., 2018) | 63.8 | 37.4 | **62.7** | 54.4 | 24.5 | 52.9 | 58.2 | 20.6 | 22.5 |
| R-VGAE [IEEE TKDE'22] (Mrabah et al., 2022) | 71.3 | 49.8 | 48.0 | 44.9 | 19.9 | 12.5 | 69.2 | 30.3 | 30.9 |
| **Q-VGAE (Ours)** | 72.7 | **58.6** | 49.6 | 66.1 | **47.4** | 50.2 | 64.3 | 32.6 | 28.0 |
| VGAECD-OPT [Entropy'20] (Choong et al., 2020) | 27.2 | 37.3 | 22.0 | 51.8 | 25.1 | 15.5 | 32.2 | 25.0 | 26.1 |
| Mod-Aware VGAE [NN'22] (Salha-Galvan et al., 2022) | 67.1 | 52.4 | 44.8 | 51.8 | 25.1 | 15.5 | - | 30.0 | 29.1 |
| GMM-VGAE [AAAI'20] (Hui et al., 2020) | 71.9 | 53.3 | 48.2 | 67.5 | 40.7 | 42.4 | 71.1 | 29.9 | 33.0 |
| R-GMM-VGAE [IEEE TKDE'22] (Mrabah et al., 2022) | **76.7** | 57.3 | **57.9** | 68.9 | 42.0 | 43.9 | **74.0** | 33.4 | **37.9** |
| **Q-GMM-VGAE (Ours)** | 76.2 | **58.7** | 56.3 | **72.7** | **47.4** | **48.8** | 69.0 | **34.8** | 34.0 |

Table 2: **Comparison of all methods on attributed datasets.** We classify the baselines into three primary groups: the first includes traditional clustering algorithms and GNN-free methods, serving as relevant baselines for our Q-MAGC method. The second category compares GNN-based methods with Q-GCN, and the last, comprising the most performant methods, consists of VGAE-based baselines for Q-VGAE and Q-GMM-VGAE.

## 5.5 Ablation Studies

**Comparison of running times**

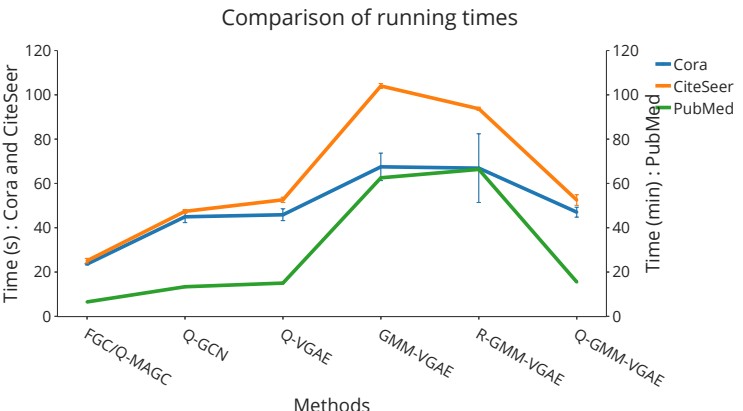

Figure 2: Comparison of running times of methods. scale for PubMed (right axis): mins, scale for others (left axis) : secs. Q-VGAE and Q-GMM-VGAE achieve better performance while needing 75% less runtime. Q-MAGC achieves ≈90% of this performance while needing less than half the runtime.

In Figure 2, we compare the running times of our method with other baselines. Our method consistently takes less than half the time across all datasets. Notably, on PubMed (large dataset), state-of-the-art methods GMM-VGAE and R-GMM-VGAE (unmodified) require approximately 60 minutes for clustering,

| Method | Brazil | | | Europe | | | USA | | |
|---|---|---|---|---|---|---|---|---|---|
| | ACC ↑ | NMI ↑ | ARI ↑ | ACC ↑ | NMI ↑ | ARI ↑ | ACC ↑ | NMI ↑ | ARI ↑ |
| GAE [NeurIPS'16] | 62.6 | 37.8 | 30.8 | 47.6 | 19.9 | 12.7 | 43.9 | 13.6 | 11.8 |
| DGI [ICLR'19] | 64.9 | 31.0 | 30.4 | 48.6 | 16.1 | 12.3 | **52.2** | 22.9 | **21.7** |
| GIC [PAKDD'21] | 40.5 | 23.5 | 14.1 | 40.4 | 9.4 | 6.2 | 49.7 | 22.1 | 19.9 |
| DAEGC [AAAI'19] | 71.0 | **47.4** | 41.2 | 53.6 | 30.9 | 23.3 | 46.4 | **27.2** | 18.4 |
| **Q-GCN (Ours)** | 51.1 | 31.9 | 23.7 | 45.5 | 30.8 | 25.1 | 43.8 | 19.1 | 14.8 |
| VGAE [NeurIPS'16] | 64.1 | 38.0 | 30.7 | 49.9 | 23.5 | 16.7 | 45.8 | 23.1 | 15.7 |
| **Q-VGAE (Ours)** | 50.1 | 35.0 | 19.8 | 46.6 | 19.5 | 17.5 | 46.2 | 19.5 | 16.9 |
| GMM-VGAE [AAAI'20] | 70.2 | 46.0 | 41.9 | 53.1 | 31.1 | 24.4 | 48.1 | 21.9 | 13.2 |
| R-GMM-VGAE [IEEE TKDE'22] | **73.3** | 45.6 | **42.5** | **57.4** | 31.4 | **25.8** | 50.8 | 23.1 | 15.3 |
| **Q-GMM-VGAE (Ours)** | 68.4 | 46.0 | 42.4 | 47.9 | **32.2** | 23.5 | 46.6 | 23.1 | 13.1 |

Table 3: Comparison of all methods on non-attributed datasets.

whereas our Q-GMM-VGAE delivers superior performance in under 15 minutes, representing a 75% reduction. Additionally, Q-MAGC runs even faster, completing in just 6 minutes on PubMed, and achieves approximately 90% of the performance.

**Modularity Metric Comparison with FGC and DMoN**

| | Cora | | | CiteSeer | | | PubMed | | |
|---|---|---|---|---|---|---|---|---|---|
| | $\mathcal{C}$ ↓ | $\mathcal{Q}$ ↑ | NMI ↑ | $\mathcal{C}$ ↓ | $\mathcal{Q}$ ↑ | NMI ↑ | $\mathcal{C}$ ↓ | $\mathcal{Q}$ ↑ | NMI ↑ |
| DMoN | 12.2 | **76.5** | 48.8 | 5.1 | **79.3** | 33.7 | 17.7 | **65.4** | 29.8 |
| FGC | 58.4 | 25 | 23.1 | 41.6 | 41.1 | 31 | 21.6 | 44.1 | 20.5 |
| Q-MAGC | 13.3 | 72.5 | 51.7 | 16.8 | 64.9 | 40.16 | 26 | 40.3 | 28.1 |
| Q-GCN | 13.6 | 73.3 | 58.3 | 5.8 | 74.5 | 46.7 | **8.27** | 55 | 31.5 |
| VGAE | 17.6 | 60.8 | 38.1 | 12.8 | 55.8 | 21 | 13.5 | 45.8 | 26.9 |
| Q-VGAE | **9.5** | 71.5 | **58.4** | **4.6** | 72.4 | **47.3** | 9.4 | 52.12 | **31.8** |

Table 4: Comparison of modularity and conductance at the best NMI with FGC and DMoN. Note that DMoN is optimizing only modularity, whereas we are optimizing other important terms as well, as mentioned in Equation 5, and thus gain a lot on NMI by giving up a small amount of modularity, making us closer to the ground truth.

We treat modularity as a metric and measure the gains observed in modularity over other baselines on the Cora, CiteSeer, and PubMed datasets. We report two types of graph-based metrics: modularity $\mathcal{Q}$ and conductance $\mathcal{C}$, which do not require labels. Conductance measures the fraction of total edge volume pointing outside the cluster, with $\mathcal{C}$ being the average conductance across all clusters, where a lower value is preferred.

From Table 4, we observe that although DMoN (Tsitsulin et al., 2023) achieves the highest modularity, our method attains significantly higher NMI. For CiteSeer, we achieve a 40% improvement in NMI with only an 8% decrease in modularity, positioning us closer to the ground truth. Additionally, our methods outperform their foundational counterparts, with Q-MAGC outperforming FGC, and Q-VGAE outperforming VGAE. Although modularity is a valuable metric to optimize, the maximum modularity labeling of a graph does not always correspond to the ground truth labeling. Therefore, it is crucial to include the other terms in our formulation. While optimizing modularity helps approach the optimal model parameters (where NMI = 1), it can deviate slightly. The additional terms in our formulation correct this trajectory, improving results.

**Importance of and Evolution of different loss terms** We analyze the evolution of the different loss terms during training and empirically show in Figure 3b that the value of each term converges and no terms are counteracting. We provide more details in Appendix N. Additionally, we measure the impact of each term separately by removing terms from the loss one by one as shown in Figure 3a. This shows that the synergy of these terms when combined is essential for clustering.

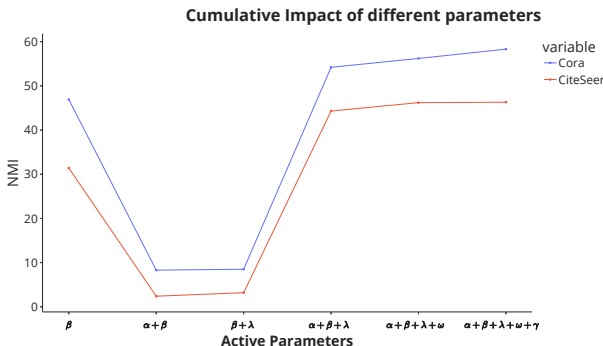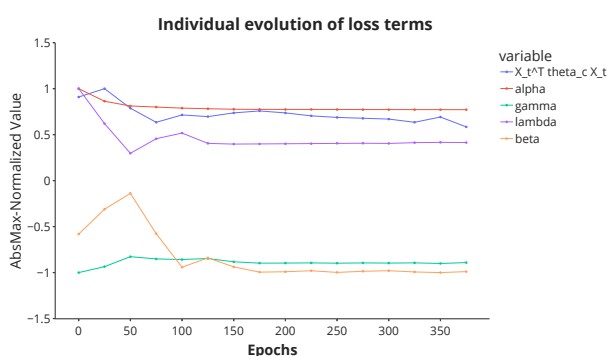

Figure (3a): Impact of active parameters on clustering performance. All terms in the loss Equation 5 are represented by their parameters, for example the modularity term by $\beta$. $\omega$ represents the non-parameterized term.

Figure (3b): Evolution of the different loss terms while training on the Cora dataset. Each series has been normalized by its absolute maximum value to allow for observation of convergence behavior. All terms show convergence except for $\gamma$ which remains almost constant.

Figure 3: Ablations on independent loss terms. Each term is represented by its weight parameter.

Also, in Appendix subsection K.2 we conduct a sensitivity study, and we find that $||CX_C - X||_F^2$ is the most sensitive to a change in its weight $\alpha$, followed by the terms related to $\gamma$, $\beta$ and then $\lambda$. This makes sense because if that constraint(relaxation) is not being met, then $C$ would have errors. Even though some of the terms do the heavy lifting, the other regularization terms do contribute to performance and more importantly, change the nature of $C$ : The smoothness term corresponds to smoothness of signals in the graph being transferred to the coarsened graph which encourages local "patches"/groups of $C$ to belong to the same cluster. The term $\gamma$ ensures that the coarsened graph is connected - i.e. preserving inter-cluster relations, which simple contrastive methods destroy; this affects $C$ by making it so that $\Theta_C$ has minimal multiplicity of 0-eigenvalues.

**Visualization of the latent space**

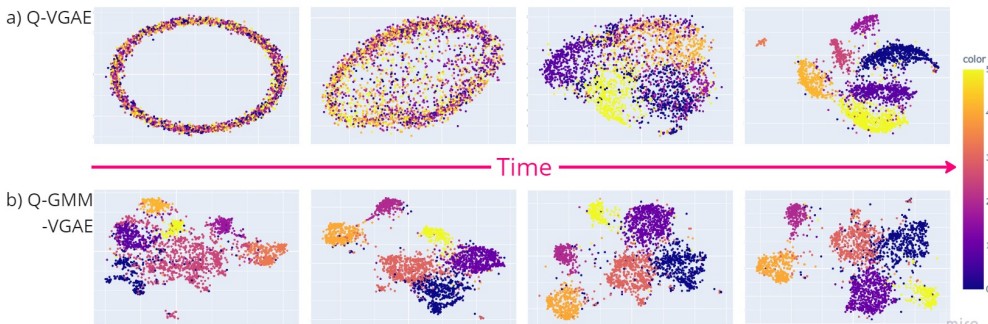

Figure 4: Evolution of the latent space of a) Q-VGAE and b) Q-GMM-VGAE over time for Cora. Colors represent clusters.

In Figure 4, we visualize how the latent space of the Q-VGAE and Q-GMM-VGAE changes over time for the Cora dataset. Plots for the rest of the datasets can be found in the Appendix O. We use UMAP (Uniform Manifold Approximation and Projection) (McInnes et al., 2018) for dimensionality reduction. Q-VGAE starts from random initialization whereas Q-GMM-VGAE is initialized with the weights learnt from GMM-VGAE. Both embeddings improve as training progresses. This shows that Q-GMM-VGAE learns a better representation i.e. clusterable embeddings which improve over time.

## 6 Discussion

**Performance.** Our methods consistently outperform their counterparts. Q-MAGC demonstrates significant superiority over traditional algorithms and GNN-free approaches because of the inclusion of modularity, smoothness, and connectedness terms in our objective. To compete with GNN-based methods, we leverage their powerful message-passing mechanisms. Q-GCN surpasses other GCN-based methods, such as AGE and DCRN. For the same reasons, Q-VGAE achieves better results than other VGAE-based methods.

**Efficiency.** Our methods are efficient in both complexity and implementation, as demonstrated in the Complexity Analysis 1 and the Ablation - Comparison of running times.

So, we conclude that Q-MAGC is well suited for light/edge applications where efficiency and reproducibility is key, since it is deterministic.

The Q-GCN architecture assigns the role of both embedding the graph and predicting the assignment matrix to the same network, so it is understandably less performant.

In most cases, Q-VGAE should serve better, considering the extra GMM pretraining that is required for the GMM-VGAE architecture. Depending on underlying data distribution, Q-GMM-VGAE might work better if the distribution is very complex and would help from separate priors being learnt for each cluster.

**Limitations.** If the modularity of a graph, calculated from the ground truth labeling, is low, our method may not perform optimally and can be slower to converge since maximizing modularity in such cases is not ideal. However, our method still manages to match or surpass state-of-the-art methods on the Airports dataset, where all graphs exhibit low modularity based on ground truth labels. This is due to the other terms in our weighted loss function, which become the primary contributors to performance.

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

# Appendix

# A    Implementation

The implementations for all the experiments can be found at `https://github.com/plutonium-239/MAGC`.

We have extensively used the PyTorch(Paszke et al., 2019) and PyTorch Geometric(Fey & Lenssen, 2019) libraries in our implementations and would like to thank the authors and developers.

# B    Literature Survey

In this section, we review relevant existing works and highlight their limitations, thereby motivating the need for an improved clustering formulation.

**Graph Clustering via Coarsening.** Graph coarsening can be extended to graph clustering by reducing the size of the coarsened graph to the number of classes ($k$). However, in most graphs, the number of classes is very small, and reducing the graph to this extent may lead to poor clustering quality. DiffPool (Ying et al., 2018) learns soft cluster assignments at each layer of the GNN and optimizes two additional losses, an entropy to penalize the soft assignments and a link prediction based loss. Next, SAGPool (Lee et al., 2019) calculates attention scores and node embeddings to determine the nodes that need to be preserved or removed. Top-k (Gao & Ji, 2019) also works by sparsifying the graph with the learned weights. MinCutPool (Bianchi et al., 2020) formulates a differentiable relaxation of spectral clustering via pooling. However, Tsitsulin et al. (2023) show that MinCutPool's orthogonal regularization dominates over the clustering objective and the objective is not optimized. Some disadvantages of these methods are instability and computational complexity in the case of SAGPool and DiffPool and convergence in MinCutPool. To address these challenges, we need a better loss function to perform the clustering task effectively.

**Deep Graph Clustering.** Previous literature can be classified based on contrastive and non-contrastive methods. On the non-contrastive side, Pan et al. (2018) proposed ARGA and ARVGA, enforcing the latent representations to align to a prior using adversarial learning. By utilizing an attention-based graph encoder and a clustering alignment loss, Wang et al. (2019) propose DAEGC. Liu et al. (2022) design the DCRN model to alleviate representation collapse by a propagation regularization term minimizing the Jensen Shannon Divergence (JSD) between the latent and its product with normalized $A$. Contrastive methods include AGE (Cui et al., 2020) which builds a training set by adaptively selecting node pairs that are highly similar or dissimilar after filtering out high-frequency noises using Laplacian smoothing. Zhao et al. (2021) propose GDCL to correct the sampling bias by choosing negative samples based on the clustering label.

VGAEs (Kipf & Welling, 2016a) are an increasingly popular class of GNNs that leverage variational inference (Kingma & Welling, 2014) for learning latent graph representations in unsupervised settings. They reconstruct the adjacency matrix after passing the graph through an encoder-decoder architecture. Many attempts have been made to use VGAEs with k-means on latent embeddings, but it has been unsuitable for clustering. This is primarily because embedded manifolds obtained from VGAEs are curved and must be flattened before any clustering algorithms using Euclidean distance are applied. Refer to Appendix P for a detailed explanation. VGAEs only use a single Gaussian prior for the latent space, whereas clustering requires the integration of meta-priors. Additionally, the inner-product decoder fails to capture locality and cluster information in the formed edges. Several clustering-oriented variants of VGAEs (Mrabah et al., 2022; Hui et al., 2020) have been developed that overcome most of these challenges. GMM-VGAE (Hui et al., 2020) partitions the latent space using a Gaussian Mixture Model and assigns a separate prior for each cluster to better model complex data distributions. Despite the improvement in performance, it's inner-product decoder cannot capture locality information.

**Modularity Maximization.** Various heuristic algorithms have been established that solve the NP-hard problem of modularity maximization including sampling, simulated annealing (Guimerà & Nunes Amaral, 2005; Guimerà & Amaral, 2005), and greedy algorithms (Louvain/Leiden) (Newman, 2004; Blondel et al., 2008). These algorithms require intensive compute and don't use node features. Modularity maximization using GNNs has also garnered attention recently. DMoN (Tsitsulin et al., 2023) optimizes only for modularity with a collapse regularization to prevent the trivial solution, but offers no theoretical guarantees about convergence. Modularity-Aware GAEs and VGAEs (Salha-Galvan et al., 2022) use a prior membership matrix

using Louvain algorithm and optimize for modularity using an RBF kernel as a proxy for same-community assignment. DGCLUSTER (Bhowmick et al., 2023) is a semi-supervised method that makes use of either a subset of labels or pairwise memberships as Auxiliary Information coupled with modularity.

## C  Convexity of terms in the optimization objective 5

When $X_C$ is kept constant, $\mathcal{L}_{MAGC}$ gets reduced to:

$$\min_C f(C) = \text{tr}(X_C^T C^T \Theta C X_C) + \frac{\alpha}{2} \|CX_C - X\|_F^2 - \frac{\beta}{2e}\text{tr}(C^T BC) \tag{15}$$

$$- \gamma \log\det(C^T \Theta C + J) + \frac{\lambda}{2} \|C^T\|_{1,2}^2$$

$$\text{subject to } \mathcal{C} \in \mathcal{S}_c 1 \text{ where, } J = \frac{1}{k}\mathbf{1}_{k \times k} \tag{16}$$

The term $\text{tr}(X_C^T C^T \Theta C X_C)$ is convex function in $C$. This result can be derived easily using Cholesky Decomposition on the positive semi-definite matrix $\Theta$ (i.e. $\Theta = L^T L$):

$$\text{tr}(X_C^T C^T \Theta C X_C) = \text{tr}(X_C^T C^T L^T L C X_C) = \text{tr}((LCX_C)^T LCX_C) = \|LCX_C\|_F^2$$

Frobenius norm is a convex function, and the simplified expression is linear in C. Hence we can deduce that the $\text{tr}(.)$ term is convex in C. The terms $\|CX_C - X\|_F^2$ and $\|C^T\|_{1,2}^2$ are convex because Frobenius norm and $l_{1,2}$ norm are convex in C.

For proving the convexity of $-\log\det(C^T \Theta C + J)$ we restrict function to a line. We define a function $g$:

$$g(t) = f(z + tu)\text{where, } t \in dom(g), z \in dom(f), u \in \mathbb{R}^n.$$

A function $f: \mathbb{R}^n \to \mathbb{R}$ is convex if $g: \mathbb{R} \to \mathbb{R}$ is convex.

The graph Laplacian matrix of the coarsened graph ($\Theta_c$) is symmetric and positive semi-definite having a rank of k-1. To convert $\Theta_c$ to positive definite matrix, we add a rank 1 matrix $J = \frac{1}{k}\mathbf{1}_{k \times k}$. ($\Theta_c + J = L^T L$)

$$f(L) = -\log\det(C^T \Theta C + J) = -\log\det(L^T L) \tag{17}$$

Now substituting L = Z + tU in the above equation.

$$g(t) = -\log\det((Z + tU)^T(Z + tU))$$
$$= -\log\det(Z^T Z + t(Z^T U + U^T Z) + t^2 U^T U)$$
$$= -\log\det(Z^T(I + t(UZ^{-1} + (UZ^{-1})^T) + t^2(Z^{-1})^T U^T U Z^{-1})Z)$$
$$\text{substituting } P = UZ^{-1}$$
$$= -(\log\det(Z^T Z) + \log\det(I + t(P + P^T) + t^2 P^T P))$$
$$\text{Eigenvalue decomposition of} P = Q\Lambda Q^T \text{and} QQ^T = I$$
$$= -(\log\det(Z^T Z) + \log\det(QQ^T + 2tQ\Lambda Q^T + t^2 Q\Lambda^2 Q^T))$$
$$= -(\log\det(Z^T Z) + \log\det(Q(I + 2t\Lambda + t^2\Lambda^2)Q^T))$$
$$= -\log\det(Z^T Z) - \sum_{i=1}^{n} \log(1 + 2t\lambda_i + t^2\lambda^2)$$

Finding double derivative of $g(t)$:

$$g''(t) = \sum_{i=1}^{n} \frac{2\lambda_i^2(1 + t\lambda_i)^2}{(1 + 2t\lambda_i + t^2\lambda_i^2)^2} \tag{18}$$

Since $g"(t) \geq 0 \forall\ t \in \mathbb{R}$, $g(t)$ is a convex function in $t$. This implies $f(L)$ is convex in $L$. We know that, $C^T \Theta C + J = L^T L$ so,

$$L = \Theta^{\frac{1}{2}} C + \frac{1}{\sqrt{kp}} \mathbf{1}_{p \times k}$$

Since $L$ is linear in $C$ and $f(L)$ is convex in $L$, $-\log \det(C^T \Theta C + J)$ is convex in C.

## D    Optimal Solution of Optimization Objective in Equation 6

We first show that the function $f(C)$ is $L - Lipschitz$ continuous gradient function with $L = \max(L_1, L_2, L_3, L_4, L_5)$, where $L_1, L_2, L_3, L_4, and L_5$ are the Lipschitz constants of $\mathrm{tr}(X_C^T C^T \Theta C X_C), -\frac{\beta}{2e} \mathrm{tr}(C^T B C), -\gamma \log \det(C^T \Theta C + J), \frac{\alpha}{2} \|C X_C - X\|_F^2, and \frac{\lambda}{2} \|C^T\|_{1,2}^2$.

For the $\mathrm{tr}(X_C^T C^T \Theta C X_C)$ term, we apply triangle inequality and employ the property of the norm of the trace operator: $||tr|| = \sup\limits_{M \neq 0} \frac{|tr(M)|}{||M||_F}$.

$$\begin{aligned}
&|tr(X_C^T C_1^T \Theta C_1 X_C) - tr(X_C^T C_2^T \Theta C_2 X_C)| \\
&= |tr(X_C^T C_1^T \Theta C_1 X_C) - tr(X_C^T C_2^T \Theta C_1 X_C) + tr(X_C^T C_2^T \Theta C_1 X_C) - tr(X_C^T C_2^T \Theta C_2 X_C)| \\
&\leq |tr(X_C^T C_1^T \Theta C_1 X_C) - tr(X_C^T C_2^T \Theta C_1 X_C)| + |tr(X_C^T C_2^T \Theta C_1 X_C) - tr(X_C^T C_2^T \Theta C_2 X_C)| \\
&\leq ||tr|| ||X_C^T (C_1 - C_2)^T \Theta C_1 X_C||_F + ||tr|| ||X_C^T C_2^T \Theta (C_1 - C_2) X_C||_F \\
&\leq ||tr|| ||X_C||_F ||\Theta|| ||C_1 - C_2||_F (||C_1||_F + ||C_2||_F) \quad \text{(Frobenius Norm Property)} \\
&\leq 2\sqrt{p} ||tr|| ||X_C||_F ||\Theta|| ||C_1 - C_2||_F \quad (||C_1||_F = ||C_2||_F = \sqrt{p}) \\
&\leq L_1 ||C_1 - C_2||_F
\end{aligned}$$

The term $\frac{\alpha}{2} \|C X_C - X\|_F^2$ can be written as:

$$\begin{aligned}
&\frac{\alpha}{2} tr((C X_C - X)^T (C X_C - X)) \\
&= \frac{\alpha}{2} tr(X_C^T C^T C X_C - X^T C X_C + X^T X - X_C^T C^T X) \\
&= \frac{\alpha}{2} (tr(X_C^T C^T C X_C) - tr(X^T C X_C) + tr(X^T X) - tr(X_C^T C^T X))
\end{aligned}$$

All the sub-terms except $tr(X^T X)$ (constant with respect to C) obtained in the expression will follow similar proofs to $\mathrm{tr}(X_C^T C^T \Theta C X_C)$.

Next we consider the modularity term:

$$\begin{aligned}
&|tr(C_1^T B C_1) - tr(C_2^T B C_2)| & (19) \\
&= |tr(C_1^T B C_1) - tr(C_2^T B C_1) + tr(C_2^T B C_1) - tr(C_2^T B C_2)| & (20) \\
&\leq |tr(C_1^T B C_1) - tr(C_2^T B C_1)| + |tr(C_2^T B C_1) - tr(C_2^T B C_2)| & (21) \\
&\leq ||tr|| ||(C_1 - C_2)^T B C_1||_F + ||tr|| ||(C_1 - C_2)^T B C_2||_F & (22) \\
&\leq ||tr|| ||B|| ||C_1 - C_2||_F (||C_1||_F + ||C_2||_F) \quad \text{(Frobenius Norm Property)} & (23) \\
&\leq L_2 ||C_1 - C_2||_F & (24)
\end{aligned}$$

The Lipschitz constant for $-\gamma \log \det(C^T \Theta C + J)$ is linked to the smallest non-zero eigenvalue of the coarsened Laplacian matrix $(\Theta_c)$ and is bounded by $\frac{\delta}{(k-1)^2}$ (Rajawat & Kumar, 2017), where $\delta$ is the minimum non-zero weight of $G_c$.

$tr(\mathbf{1}^T C^T C \mathbf{1})$ can be proved to be $L_5 - Lipschitz$ like the modularity and Dirichlet energy (smoothness) terms. This concludes the proof.

The majorized problem for L-Lipschitz and differentiable functions can now be applied. The Lagrangian of the majorized problem, 10 is:

$$\mathcal{L}(C, X_C, \mu) = \frac{1}{2}C^T C - C^T C_{\text{un}}^{(t+1)} - \mu_1^T C + \mu_2^T \left[ \left\| C_1^T \right\|_2^2 - 1, \cdots, \left\| C_i^T \right\|_2^2 - 1, \cdots, \left\| C_p^T \right\|_2^2 - 1 \right]^T \quad (25)$$

where $\mu = [\mu_1, \mu_2]$ are the dual variables and $C_{\text{un}}^{(t+1)} = \left( C^{(t)} - \frac{1}{L}\nabla f(C^{(t)}) \right)^+$ is the unnormalized $C$ after $t^{th}$ majorizing iteration.

The corresponding KKT conditions (w.r.t $C$) are:

$$C - A - \mu_1 + 2[\mu_{2_o}C_0^T, \cdots, \mu_{2_i}C_i^T, \cdots, \mu_{2_p}C_p^T] = 0 \quad (26)$$

$$\mu_2^T \left[ \left\| C_1^T \right\|_2^2 - 1, \cdots, \left\| C_i^T \right\|_2^2 - 1, \cdots, \left\| C_p^T \right\|_2^2 - 1, \right]^T = 0 \quad (27)$$

$$\mu_1^T C = 0 \quad (28)$$

$$\mu_1 \geq 0 \quad (29)$$

$$\mu_2 \geq 0 \quad (30)$$

$$C \geq 0 \quad (31)$$

$$\left\| [C^T]_i \right\|_2^2 \leq 1 \; \forall i \quad (32)$$

The optimal solution to these KKT conditions is the normalized version of $C_{\text{un}}^{(t+1)}$:

$$C^{(t+1)} = \frac{(C_{\text{un}}^{(t+1)})^+}{\sum_i \left\| [C_{\text{un}}^{(t+1)^T}]_i \right\|_2} \quad (33)$$

## E   Proof of Theorem 1 (Convergence)

In this section, we prove that the sequence $\{C^{t+1}, X_C^{t+1}\}$ generated by Algorithm 1 converges to the set of Karush–Kuhn–Tucker (KKT) optimality points for Problem 5.

The Lagrangian of Problem 5 comes out to be:

$$\mathcal{L}(C, X_C, \mu) = \text{tr}(X_C^T C^T \Theta C X_C) + \frac{\alpha}{2}\left\| C X_C - X \right\|_F^2 - \frac{\beta}{2e}\text{tr}(C^T B C) \quad (34)$$

$$- \gamma \log \det(C^T \Theta C + J) + \frac{\lambda}{2}\left\| C^T \right\|_{1,2}^2 - \mu_1^T C + \sum_i \mu_{2i}\left[ \left\| C_i^T \right\|_2^2 - 1 \right] \quad (35)$$

where $\mu = [\mu_1, \mu_2]$ are the dual variables.

**w.r.t.** $C$, the KKT conditions are

$$2\Theta C X_C X_C^T + \alpha(C X_C - X)X_C^T - \frac{\beta}{e}B C - 2\gamma\Theta C(C^T \Theta C + J)^{-1} \quad (36)$$

$$+\lambda C \mathbf{1}_{k \times k} - \mu_1 + 2[\mu_{2_o}C_0^T, \cdots, \mu_{2_i}C_i^T, \cdots, \mu_{2_p}C_p^T] = 0$$

$$\mu_2^T \left[ \left\| C_1^T \right\|_2^2 - 1, \cdots, \left\| C_i^T \right\|_2^2 - 1, \cdots, \left\| C_p^T \right\|_2^2 - 1 \right]^T = 0 \quad (37)$$

$$\mu_1^T C = 0 \quad (38)$$

$$\mu_1 \geq 0 \quad (39)$$

$$\mu_2 \geq 0 \quad (40)$$

$$C \geq 0 \quad (41)$$

$$\left\| [C^T]_i \right\|_2^2 \leq 1 \; \forall i \quad (42)$$

Now, $C^\infty \equiv \lim_{t \to \infty} C^t$ is found from Equation 11 as:

$$C^\infty = C^\infty + \frac{1}{L}\left(2\Theta C^\infty X_C^\infty X_C^\infty + \alpha(C^\infty X_C - X)X_C^\infty - \frac{\beta}{e}BC^\infty \right.$$
$$\left. - 2\gamma\Theta C^\infty(C^{\infty T}\Theta C^\infty + J)^{-1} + \lambda C^\infty \mathbf{1}_{k \times k}\right) \tag{43}$$

$$0 = 2\Theta C^\infty X_C^\infty X_C^\infty + \alpha(C^\infty X_C - X)X_C^\infty - \frac{\beta}{e}BC^\infty$$
$$- 2\gamma\Theta C^\infty(C^{\infty T}\Theta C^\infty + J)^{-1} + \lambda C^\infty \mathbf{1}_{k \times k} \tag{44}$$

So, for $\mu = 0$, $C^\infty$ satisfies the KKT conditions.

**w.r.t.** $X_C$, the KKT conditions are:

$$2C^T\Theta CX_C + \alpha C^T(CX_C - X) = 0 \tag{45}$$

So, $X^\infty \equiv \lim_{t \to \infty} X^t$ found from Equation 14 will satisfy this as that equation is just a rearrangement of the KKT condition.

## F Complexities of some graph clustering methods

Some GCN-*based* clustering methods:

- AGC(Zhang et al., 2019) - $\mathcal{O}(p^2nt + ent^2)$
  where $t$ is the number of iterations (within an epoch)
- R-VGAE(Mrabah et al., 2022) - $\mathcal{O}(pk^2n + (p(n+k) + e(p+k))$
- S3GC(Devvrit et al., 2022) - $\mathcal{O}(pn^2s)$
  where $s$ is the average degree
- HSAN(Liu et al., 2023b) - $\mathcal{O}(pBn)$
  they state it as $\mathcal{O}(B^2d)$ but that is only for 1 batch of size $B$ and not the whole epoch
- VGAECD-OPT(Choong et al., 2020) - $\mathcal{O}(p^2nD^L)$
  where $D$ is the size of graph filter, $l$ is the number of linear layers

## G Proof of Theorem 2 (Consistency Analysis on DC-SBMs)

### G.1 Detailed Proof of Theorem 2

*Proof.* The proof is divided into two parts. First, we demonstrate that $\mathcal{L}_{MAGC}$ can be expressed in terms of $O$ and $\pi$. Second, we verify that, in this form, $\mathcal{L}_{MAGC}$ satisfies the regularity conditions specified in Theorem 4.1 of Zhao et al. (2012). It is important to note that DC-SBMs do not consider node features, so we treat $X$ and $X_C$ as constants. We start by noting that $O = C^TAC$, i.e., the edge count matrix $O$ is the class-transformed version of the adjacency matrix $A$. This has a good intuition behind it, as $C$ is the mapping from nodes to classes, and it can also be proved easily.

$$\{C^TAC\}_{ql} = (C^T)_qAC_{:l} = \sum_{i=1}^{p}\sum_{j=1}^{p}(C^T)_{qi}A_{ij}C_{jl} = \sum_{i=1}^{p}\sum_{j=1}^{p}C_{iq}A_{ij}C_{jl} \tag{46}$$

$C_{iq}$ equals 1 only when node $i$ is in cluster $q$, and $C_{jl}$ equals 1 only when node $j$ is in cluster $l$. Therefore, this expression simplifies to the definition of $O$.

$$\{C^TAC\}_{ql} = \sum_{i=1}^{p}\sum_{j=1}^{p}A_{ij}\mathbb{1}\{y_i = q, y_j = l\} \tag{47}$$

This can be viewed as the *cluster-level adjacency matrix*, which quantifies the edges between clusters. Next, we formalize the relationship between $O$ and $C^T DC$, where $D = \text{diag}(d)$ is the degree matrix.

$$\{C^T DC\}_{ql} = \sum_{i=1}^{p}\sum_{j=1}^{p} C_{iq}D_{ij}C_{jl} = \sum_{i=1}^{p} C_{iq}d_i C_{il} \quad [D_{ij} = 0 \text{ if } i \neq j] \tag{48}$$

Since the rows of $C$ are orthonormal, $C_{iq}C_{il} = 1$ if and only if $q = l$. Thus, we obtain the following expression for the diagonal entries of $C^T DC$:

$$\{C^T DC\}_{qq} = \sum_{i=1}^{p} C_{iq}^2 d_i = \sum_{i=1}^{p} C_{iq}d_i \quad \left[\because C_{ij} \in \{0, 1\}\right] \tag{49}$$

Now, we consider $\sum_{l=1}^{k}\{C^T AC\}_{ql}$

$$= \sum_{i=1}^{p}\sum_{j=1}^{p}\left(C_{iq}A_{ij}\sum_{l=1}^{k}C_{jl}\right) = \sum_{i=1}^{p}\sum_{j=1}^{p} C_{iq}A_{ij}\cdot 1 = \sum_{i=1}^{p} C_{iq}d_i \tag{50}$$

Thus, we showed that $C^T DC = \text{diag}(\sum_{l=1}^{k}[C^T AC]_k)$, which intuitively represents the *cluster-level degree matrix* because it is the (diagonal of) row/column-sum of the $C^T AC$ or cluster-level adjacency matrix. We now express the Laplacian of the coarsened graph as a function of $O$, $\Theta_C(O) = C^T\Theta C = C^T(D - A)C = \text{diag}(\sum_{l=1}^{k} O_l) - O$. This implies that the terms $\text{tr}(X_C^T C^T\Theta C X_C)$, $\text{tr}(C^T BC)$, and $\log\det(C^T\Theta C + J)$ in our clustering objective are functions of $\Theta_C(O)$. Using Theorem 4.1 from Zhao et al. (2012), we conclude that $\mathcal{L}_{\text{MAGC}}$ must be uniquely minimized at any point $(y^*, A^*)$ s.t. $\mathbb{E}_p[\pi(y_p)] = \pi(y^*)$ and $\mathbb{E}_p[A^{(p)}] = A^*$. The Laplacian $\Theta$ can be written as a function of the adjacency $\Theta_{ij} = \sum_j A_{ij}^* - A_{ij}^*$ (assuming no self-loops).

For the modularity term, this has been proven in Theorem 3.1 of Zhao et al. (2012). For the constraint relaxation term $\frac{1}{2}\|CX_C - X\|_F^2$, we see that $X_C$ is a function of only $C(y^*)$ and $\Theta(A^*)$ from it's update rule equation 14, so we get $\text{term}_\alpha(y^*, \Theta) = \frac{1}{2}\|C(y^*)X_C(y^*, A^*) - X\|_F^2$. $\text{tr}(X_C C^T\Theta C X_C)$ is already just a function of $X_C, C, \Theta$ which we have shown above to be functions of only $y^*, A^*$. By the same logic, $-\log\det(C^T\Theta C + J)$ is also a function of only $y^*, A^*$ ($J$ is a constant $= \frac{1}{k}\mathbf{1}_{k\times k}$). Thus, $\mathcal{L}_{\text{MAGC}}$ can be written as required above.

Next, we compute the *population version* of $\mathcal{L}_{MAGC}$.

$$F(H(S), h(S)) = \mathbb{E}[-\mathcal{L}_{MAGC}|C, t] = \underbrace{f_1(H(S), h(S))}_{\text{smoothness}} + \underbrace{f_2(H(S), h(S))}_{\text{log-determinant}} + \underbrace{f_3(H(S), h(S))}_{\text{modularity}} \tag{51}$$

where, $f_1(H(S), h(S)) = \mu_p\text{tr}(X_C^T[H(S) - \text{diag}(\sum_{j=1}^{k} H(S)_{ij})]X_C)$ \hfill (52)

$$f_2(H(S), h(S)) = \frac{\mu_p}{b}\text{tr}\left(H(S) - \text{diag}(\sum_{j=1}^{k} H(S)_{ij})\right) - \frac{1}{b} + k - k\log b \tag{53}$$

$$f_3(H(S), h(S)) = \frac{\text{tr}(H(S))}{\tilde{P}_0} - \frac{\sum_{i=1}^{k}(\sum_{j=1}^{k} H(S)_{ij})^2}{\tilde{P}_0^2} \tag{54}$$

$$\text{and, } \tilde{P}_0 = \sum_{ab}\tilde{\pi}_a\tilde{\pi}_b P_{ab} \quad; \quad \tilde{\pi}_a = \sum_u x_u\Pi_{au} \text{ with } \sum_a\tilde{\pi}_a = 1, \text{ since } \mathbb{E}[t_i] = 1. \tag{55}$$

For brevity, we denote $F(H(S), h(S))$ by $F(S)$. Now for the second part of the proof, it is easy to prove that $F(S)$ satisfies the regularity conditions stated in Lemma 1. The detailed proof is presented in the following subsections. This concludes the proof that $\mathcal{L}_{MAGC}$ is strongly and weakly consistent under the DC-SBM. $\qquad\square$

### G.2 Proof of Theorem 2: Conditions for Consistency

As previously defined in Theorem 2 $O = C^T AC$ and

$$\frac{1}{\mu_p}\mathbb{E}[O|C, t] = H(S)$$

We need to find "*population version*" of the loss function in terms of $H(S)$.

$$\mathbb{E}[O|C,t] = \mathbb{E}[C^T A C|C,t] = \mu_p H(S) \tag{56}$$

$$\mathbb{E}[C^T D C|C,t] = \mathbb{E}[C^T \mathrm{diag}(\sum_{j=1}^{k} A_{ij})C|C,t] = \mathbb{E}[\mathrm{diag}(\sum_{j=1}^{k} O_{ij})|C,t] \tag{57}$$

$$= \mu_p \mathrm{diag}(\sum_{j=1}^{k} H_{ij}) \tag{58}$$

So, for $\mathrm{tr}(X_C^T C^T \Theta C X_C)$

$$
\begin{aligned}
\mathbb{E}[\mathrm{tr}(X_C^T C^T \Theta C X_C)|C,t] \quad &= \mathrm{tr}(\mathbb{E}[X_C^T C^T \Theta C X_C|C,t]) \\
= \mathrm{tr}(X_C^T \mathbb{E}[C^T \Theta C|C,t] X_C) &= \mathrm{tr}(X_C^T \mathbb{E}[C^T D C - C^T A C|C,t] X_C) \\
= \mathrm{tr}(X_C^T [\mu_p \mathrm{diag}(\sum_{j=1}^{k} H_{ij}) \quad &- \mu_p H(S)] X_C) \\
= \mu_p \mathrm{tr}(X_C^T [\mathrm{diag}(\sum_{j=1}^{k} H_{ij}) \quad &- H(S)] X_C)
\end{aligned}
\tag{59}
$$

Next, we have $\frac{1}{2e}\mathrm{tr}(C^T B C)$, which has already been solved in the paper Zhao et al. (2012) in their Appendix (Page 25 of the full document).

$$\frac{1}{2e}\mathbb{E}[\mathrm{tr}(C^T B C)] = \sum_{k}\left(\frac{H_{kk}}{\tilde{P}_0} - \left(\frac{H_k}{\tilde{P}_0}\right)^2\right)$$

For $\log\det(C^T \Theta C + J)$, where $J = \frac{1}{k}\mathbb{1}_{k\times k}$,

We can write $\log(\det(C^T \Theta C + J)) = \mathrm{tr}(\log(C^T \Theta C + J))$ since, $\det(A) = e^{\mathrm{tr}(\log(A))}$.

$$Z = C^T \Theta C + J = V\Lambda V^{-1} \tag{60}$$

$$\text{So,} \tag{61}$$

$$\mathrm{tr}(\log(V\Lambda V^{-1})) = \mathrm{tr}(V\log(\Lambda)V^{-1}) \tag{62}$$

$$\log\Lambda = \log(bI) + \log\left(I + \frac{\Lambda}{b} - I\right) \tag{63}$$

Using the first-order Taylor expansion of $\log(I + X) = X$, we need to choose $b$ such that

$$l = \left\|\frac{\Lambda}{b} - I\right\|_F < 1 \tag{64}$$

And for the expansion to be a good approximation, we need $l \to 0$. We will enforce this later in Equation 74.

$$\text{tr}(V \left( \log(bI) + \log \left( I + \frac{\Lambda}{b} - I \right) \right) V^{-1}) \tag{65}$$

$$= \text{tr}(V \left( \log(bI) + \left( I + \frac{\Lambda}{b} - I \right) \right) V^{-1}) \tag{66}$$

$$= \text{tr}(V \log(bI) V^{-1}) + \frac{1}{b} \text{tr}(V \Lambda V^{-1}) - \text{tr}(I) \tag{67}$$

$$= \text{tr}(\log(bI)) + \frac{1}{b} \text{tr}(Z) - \text{tr}(I) \tag{68}$$

$$= k \log(b) + \frac{1}{b} \text{tr}(Z) - k \tag{69}$$

Finding the expectation of from Equation 63,

$$= k \log(b) + \mathbb{E} \left[ \frac{1}{b} \text{tr}(Z) \, \middle| \, c, t \right] - k \tag{70}$$

$$= k \log(b) + \frac{1}{b} \text{tr}(\mathbb{E}[Z| \, c, t]) - k \tag{71}$$

$$= k \log(b) + \frac{1}{b} \text{tr}(\mathbb{E}[C^T \Theta C + J| \, c, t]) - k \tag{72}$$

Using $\Theta = D - A$, Equation 56 and Equation 58,

$$= k \log(b) + \frac{1}{b} \text{tr} \left( \mu_p \text{diag}(\sum_{j=1}^{k} H_{ij}) - \mu_p H(S) \right) - k \tag{73}$$

which is a linear function in $H(S)$.

For the approximation to be good, we can now simplify $l$ as defined in Equation 64:

$$l = \left\| \frac{\Lambda}{b} - I \right\|_F \tag{74}$$

$$l = \left\| \begin{matrix} \frac{1}{b}\Lambda_{11} - 1 & \frac{1}{b}\Lambda_{12} & \cdots & \frac{1}{b}\Lambda_{1k} \\ \frac{1}{b}\Lambda_{21} & \frac{1}{b}\Lambda_{22} - 1 & \cdots & \frac{1}{b}\Lambda_{2k} \\ \vdots & \vdots & \ddots & \vdots \\ \frac{1}{b}\Lambda_{k1} & \frac{1}{b}\Lambda_{k2} & \cdots & \frac{1}{b}\Lambda_{kk} - 1 \end{matrix} \right\|_F \tag{75}$$

Writing out this norm, we get a quadratic expression in $\frac{1}{b}$:

$$l = (\sum_{i=1}^{k}\sum_{j=1}^{k}\lambda_{ij}^2)\frac{1}{b^2} - 2(\sum_{u=1}^{k}\lambda_{ii})\frac{1}{b} + k \tag{76}$$

$$\text{or concisely, } l = \|\Lambda\|_F^2\frac{1}{b^2} - 2\text{tr}(\Lambda)\frac{1}{b} + k \tag{77}$$

$$\text{Since, } \Lambda \text{ is a diagonal matrix,} \tag{78}$$

$$\text{tr}(\Lambda) = \sum_i \lambda_i = \text{tr}(Z) \tag{79}$$

$$\text{Also, since } \Lambda^2 \text{ is the eigenvalue matrix for } Z^2, \tag{80}$$

$$\|\Lambda\|_F^2 = \sum_i \lambda_i^2 = \text{tr}(Z^2) \tag{81}$$

$$l = \text{tr}(Z^2)\frac{1}{b^2} - 2\text{tr}(Z)\frac{1}{b} + k \tag{82}$$

Since, $l < 1 \implies l - 1 < 0 \implies l - 1 = 0$ has 2 real roots. Using simple quadratic analysis (in $\frac{1}{b}$), the discriminant $\Delta$ should be positive.

$$\Delta = 4\text{tr}(Z)^2 - 4(k-1)\text{tr}(Z^2) > 0 \tag{83}$$

$$\frac{\text{tr}(Z)^2}{\text{tr}(Z^2)} > k - 1 \tag{84}$$

$$\text{The minimum value of } l \coloneqq l_{min} \text{ occurs at} \tag{85}$$

$$b = \frac{\text{tr}(Z^2)}{\text{tr}(Z)} = \text{tr}(Z) \cdot \frac{\text{tr}(Z^2)}{\text{tr}(Z)^2} \tag{86}$$

$$l_{min} < 1 \text{ will exist when } \frac{\text{tr}(Z)^2}{\text{tr}(Z^2)} > k - 1 \tag{87}$$

$$\text{which holds for } b < \frac{\text{tr}(Z)}{k-1} \tag{88}$$

$$\text{So we can always choose } b < \frac{2k-1}{k-1} \tag{89}$$

$$\text{since } \min\text{tr}(Z) = 2k - 1 [\text{proved in 91}] \tag{90}$$

$$\min\text{tr}(C^T\Theta C + J) = \min\text{tr}(C^T\Theta C) + \text{tr}(J) \tag{91}$$
$$= \min\text{tr}(C^TDC) - \text{tr}(C^TAC) + 1 \tag{92}$$
$$= 2e - 2(e - (k-1)) + 1 \tag{93}$$
$$= 2k - 1 \tag{94}$$

Additionally, define $\tilde{\pi}_a = \sum_u x_u\Pi_{au}$ with $\sum_a \tilde{\pi}_a = 1$, since $\mathbb{E}[t_i] = 1$.

### G.3 Required Condition a): Lipschitz Continuity

Condition 1: We need to show that $|F(S_1) - F(S_2)| \leq \alpha\|S_1 - S_2\|$ (Lipschitz)

$$|F(S_1) - F(S_2)| \leq |f_1(S_1) - f_1(S_2)| + |f_2(S_1) - f_2(S_2)| + |f_3(S_1) - f_3(S_2)| \tag{95}$$

Let's first find $\|H(S_1) - H(S_2)\|_F$

$$H(S) = (Sx)P(Sx)^T \tag{96}$$

$$\|H(S_1) - H(S_2)\|_F = \|(S_1x)P(S_1x)^T - (S_2x)P(S_2x)^T\|_F \tag{97}$$

$$= \|(S_1x)P((S_1 - S_2)x)^T + ((S_1 - S_2)x)P(S_2x)^T\|_F \tag{98}$$

Next, we see $\|\text{diag}(\sum_{j=1}^k H(S_1)_{ij}) - \text{diag}(\sum_{j=1}^k H(S_2)_{ij})\|_F = \|\text{diag}(\sum_{j=1}^k (H(S_1)_{ij} - H(S_2)_{ij}))\|_F$

For the first term, $|f_1(H(S_1)) - f_1(H(S_2))|$, define $H(S)_i = \sum_{j=1}^k H(S)_{ij}$

$$= |\mu_p \text{tr}(X_C^T[H(S_1) - \text{diag}(\sum_{j=1}^k H(S_1)_{ij})]X_C) - \mu_p \text{tr}(X_C^T[H(S_2) - \text{diag}(\sum_{j=1}^k H(S_2)_{ij})]X_C)| \tag{99}$$

$$= |\mu_p \Big( \text{tr}(X_C^T[H(S_1) - H(S_2)]X_C) - \text{tr}(X_C^T \text{diag}([H(S_1)_i - H(S_2)_i])X_C) \Big)| \tag{100}$$

$$\leq |\mu_p| \Big( \Big| \text{tr}(X_C^T[H(S_1) - H(S_2)]X_C) \Big| - \Big| \text{tr}(X_C^T \text{diag}([H(S_1)_i - H(S_2)_i])X_C) \Big| \Big) \tag{101}$$

$$\leq |\mu_p| \Big( \|\text{tr}\| \Big\| X_C^T[H(S_1) - H(S_2)]X_C \Big\|_F - \|\text{tr}\| \Big\| X_C^T \text{diag}([H(S_1)_i - H(S_2)_i])X_C \Big\|_F \Big) \tag{102}$$

$$\leq |\mu_p| \Big( \|\text{tr}\|\|X_C\|^2 \Big\| H(S_1) - H(S_2) \Big\|_F - \|\text{tr}\|\|X_C\|^2 \Big\| \text{diag}([H(S_1)_i - H(S_2)_i]) \Big\|_F \Big) \tag{103}$$

Taking the first sub-term,

$$|\mu_p| \|\text{tr}\| \|X_C\|^2 \Big\| H(S_1) - H(S_2) \Big\|_F \tag{104}$$

$$= \|\text{tr}\| \|X_C\|^2 \Big\| (S_1x)P((S_1 - S_2)x)^T + ((S_1 - S_2)x)P(S_2x)^T \Big\|_F \tag{105}$$

$$\leq \|\text{tr}\| \|X_C\|^2 \|P\|_F (\|S_1x\|_F + \|S_2x\|_F) \|(S_1 - S_2)x\|_F \tag{106}$$

$$\leq \|\text{tr}\| \|X_C\|^2 \|P\|_F (\|S_1x\|_F + \|S_2x\|_F) \|t\|_F \|(S_1 - S_2)\|_F \tag{107}$$

$$= \alpha_1 \|(S_1 - S_2)\|_F \tag{108}$$

For the second sub-term, define $\tilde{S}_{ka} = \sum_u x_u S_{kau} = (Sx)_{ka}$

$$H(S)_i = \sum_{j=1}^{k} H(S)_{ij} = \sum_{as} \tilde{\pi}_s P_{as} \tilde{S}_{ia} = \sum_{asu} \tilde{\pi}_s P_{as} x_u S_{iau} \tag{109}$$

$$|H(S_1)_i - H(S_2)_i| = |\sum_{asu} \tilde{\pi}_s P_{as} x_u S_{1iau} - \sum_{asu} \tilde{\pi}_s P_{as} x_u S_{2iau}| \tag{110}$$

$$= |\sum_{asu} \tilde{\pi}_s P_{as} x_u (S_1 - S_2)_{iau}| \tag{111}$$

$$\leq \sum_{asu} |\tilde{\pi}_s P_{as} x_u (S_1 - S_2)_{iau}| \tag{112}$$

$$\leq \sum_{asu} |\tilde{\pi}_s P_{as} x_u| \cdot |(S_1 - S_2)_{iau}| \tag{113}$$

$$\leq \sum_{asu} |\tilde{\pi}_s P_{as} x_u| \cdot \sum_{au} |(S_1 - S_2)_{iau}| \tag{114}$$

$$= \alpha_1' \sum_{au} |(S_1 - S_2)_{iau}| \tag{115}$$

$$\text{So, } \|\text{diag}([H(S_1)_i - H(S_2)_i])\|_F = \sqrt{\sum_{i=1}^{k} \left( \sum_{j=1}^{k} H(S_1)_{ij} - H(S_2)_{ij} \right)^2} \tag{116}$$

$$\leq \sqrt{\sum_{i=1}^{k} \left( \alpha_1' \sum_{au} |(S_1 - S_2)_{iau}| \right)^2} \tag{117}$$

$$= \alpha_1' \|S_1 - S_2\|_{1,1,2} \tag{118}$$

For the second term, $|f_2(H(S_1)) - f_2(H(S_2))|$

$$= \frac{\mu_p}{b} \left| \text{tr}(H(S_1) - \text{diag}([H(S_1)_i])) - \text{tr}(H(S_2) - \text{diag}([H(S_2)_i])) \right| \tag{119}$$

$$= \frac{\mu_p}{b} \left| \text{tr}(H(S_1) - H(S_2)) \qquad - \text{tr}\left( \text{diag}([H(S_1)_i - H(S_2)_i]) \right) \right| \tag{120}$$

$$\leq \frac{\mu_p}{b} \left( \left| \text{tr}(H(S_1) - H(S_2)) \right| \qquad + \left| \text{tr}\left( \text{diag}([H(S_1)_i - H(S_2)_i]) \right) \right| \right) \tag{121}$$

$$\leq \frac{\mu_p}{b} \left( \|\text{tr}\| \left\| H(S_1) - H(S_2) \right\|_F \qquad + \|\text{tr}\| \left\| \text{diag}([H(S_1)_i - H(S_2)_i]) \right\|_F \right) \tag{122}$$

As shown above, $\tag{123}$

$$\leq \alpha_2 \|(S_1 - S_2)\|_F \qquad + \alpha_2' \|(S_1 - S_2)\|_{1,1,2} \tag{124}$$

For the third term, it has already been proven in Zhao et al. (2012), but we also prove it here:

$|f_3(H(S_1)) - f_3(H(S_2))|$

$$= \left| \frac{\text{tr}(H(S_1))}{\tilde{P}_0} - \frac{\sum_{i=1}^{k}(\sum_{j=1}^{k} H(S_1)_{ij})^2}{\tilde{P}_0{}^2} - \frac{\text{tr}(H(S_2))}{\tilde{P}_0} + \frac{\sum_{i=1}^{k}(\sum_{j=1}^{k} H(S_2)_{ij})^2}{\tilde{P}_0{}^2} \right| \tag{125}$$

$$= \left| \frac{\text{tr}(H(S_1) - H(S_2))}{\tilde{P}_0} - \frac{\sum_{i=1}^{k} \left[ H(S_1)_i^2 - H(S_2)_i^2 \right]}{\tilde{P}_0{}^2} \right| \tag{126}$$

$$\leq \frac{1}{\tilde{P}_0} \left| \text{tr}(H(S_1) - H(S_2)) \right| + \frac{1}{\tilde{P}_0{}^2} \left| \sum_{i=1}^{k} \left[ H(S_1)_i^2 - H(S_2)_i^2 \right] \right| \tag{127}$$

As shown above, $\tag{128}$

$$\leq \alpha_3 \|(S_1 - S_2)\|_F + \frac{1}{\tilde{P}_0{}^2} \left| \sum_{i=1}^{k} \left[ H(S_1)_i - H(S_2)_i \right] \cdot \left[ H(S_1)_i + H(S_2)_i \right] \right| \tag{129}$$

$$\leq \alpha_3 \|(S_1 - S_2)\|_F + \frac{1}{\tilde{P}_0{}^2} \sum_{i=1}^{k} \left| H(S_1)_i + H(S_2)_i \right| \cdot \sum_{i=1}^{k} \left| H(S_1)_i - H(S_2)_i \right| \tag{130}$$

$$= \alpha_3 \|(S_1 - S_2)\|_F + \alpha_3' \sum_{iau} \left| S_1 - S_2 \right|_{iau} \tag{131}$$

$$= \alpha_3 \|(S_1 - S_2)\|_F + \alpha_3' \|S_1 - S_2\|_{1,1,1} \tag{132}$$

## G.4 Required Condition b): Continuity of directional second derivative

Condition 2: $W = H(\mathbb{D})$

$$\left. \frac{\partial^2}{\partial \varepsilon^2} F(M_0 + \varepsilon(M_1 - M_0), \mathbf{t_0} + \varepsilon(\mathbf{t_1} - \mathbf{t_0})) \right|_{\varepsilon = 0^+} \tag{133}$$

$$= \left. \frac{\partial^2}{\partial \varepsilon^2} f_1(M_0 + \varepsilon(M_1 - M_0)) + \frac{\partial^2}{\partial \varepsilon^2} f_2(M_0 + \varepsilon(M_1 - M_0)) + \frac{\partial^2}{\partial \varepsilon^2} f_3(M_0 + \varepsilon(M_1 - M_0)) \right|_{\varepsilon = 0^+} \tag{134}$$

Finding the directional derivative for $f_1$,

$$f_1(M_0 + \varepsilon(M_1 - M_0)) = \mu_p \text{tr}(X_C^T [M_0 + \varepsilon(M_1 - M_0) - \text{diag}(\sum_{j=1}^{k} \left( M_0 + \varepsilon(M_1 - M_0) \right)_{ij})] X_C) \tag{135}$$

$$= \mu_p \bigg( \text{tr}(X_C^T M_0 X_C) + \varepsilon \ \text{tr}(X_C^T (M_1 - M_0) X_C) \tag{136}$$

$$- \text{tr}(X_C^T \text{diag}([\sum_{j=1}^{k} (M_0)_{ij}]) X_C) - \varepsilon \ \text{tr}(X_C^T \text{diag}([\sum_{j=1}^{k} (M_1 - M_0)_{ij}]) X_C) \bigg) \tag{137}$$

$$\frac{\partial^2}{\partial \varepsilon} f_1(M_0 + \varepsilon(M_1 - M_0)) = \mu_p \bigg( \text{tr}(X_C^T (M_1 - M_0) X_C) - \text{tr}(X_C^T \text{diag}([\sum_{j=1}^{k} (M_1 - M_0)_{ij}]) X_C) \bigg) \tag{138}$$

$$\frac{\partial^2}{\partial \epsilon^2} f_1(M_0 + \varepsilon(M_1 - M_0) = 0 \tag{139}$$

Finding the directional derivative for $f_2$,

$$f_2(M_0 + \varepsilon(M_1 - M_0)) \tag{140}$$

$$= \frac{\mu_p}{b} \text{tr}\left(M_0 + \varepsilon(M_1 - M_0) - \text{diag}(\sum_{j=1}^{k} \left(M_0 + \varepsilon(M_1 - M_0)\right)_{ij})\right) - \frac{1}{b} + k - k \log b \tag{141}$$

$$\frac{\partial}{\partial \varepsilon} f_2(M_0 + \varepsilon(M_1 - M_0)) = \frac{\mu_p}{b} \text{tr}(M_1 - M_0) - \frac{\mu_p}{b} \text{tr}\left(\text{diag}(\sum_{j=1}^{k}(M_1 - M_0)_{ij})\right) \tag{142}$$

$$\frac{\partial^2}{\partial \varepsilon^2} f_2(M_0 + \varepsilon(M_1 - M_0)) = 0 \tag{143}$$

Finding the directional derivative for $f_3$,

$$f_3(M_0 + \varepsilon(M_1 - M_0)) = \frac{\text{tr}(M_0 + \varepsilon(M_1 - M_0))}{\tilde{P}_0} - \frac{\sum_{i=1}^{k}(\sum_{j=1}^{k}(M_0 + \varepsilon(M_1 - M_0))_{ij})^2}{\tilde{P}_0^2} \tag{144}$$

$$= \frac{\text{tr}(M_0 + \varepsilon(M_1 - M_0))}{\tilde{P}_0} - \frac{\sum_{i=1}^{k}(\sum_{asu} \tilde{\pi}_s P_{as} x_u (M_0 + \varepsilon(M_1 - M_0))_{iau})^2}{\tilde{P}_0^2} \tag{145}$$

$$\frac{\partial}{\partial \varepsilon} f_3(M_0 + \varepsilon(M_1 - M_0)) =$$
$$\frac{\text{tr}(M_1 - M_0)}{\tilde{P}_0} + \sum_{i=1}^{k} 2(\sum_{asu} \tilde{\pi}_s P_{as} x_u (M_1 - M_0)_{iau}) \times \frac{\sum_{i=1}^{k}(\sum_{asu} \tilde{\pi}_s P_{as} x_u (M_0 + \varepsilon(M_1 - M_0))_{iau})^2}{\tilde{P}_0^2} \tag{146}$$

$$\frac{\partial^2}{\partial \varepsilon^2} f_3(M_0 + \varepsilon(M_1 - M_0)) = \frac{\sum_{i=1}^{k} 2(\sum_{asu} \tilde{\pi}_s P_{as} x_u (M_1 - M_0)_{iau})^2}{\tilde{P}_0^2} \tag{147}$$

Adding up these three,

$$\frac{\partial^2}{\partial \varepsilon^2} F(M_0 + \varepsilon(M_1 - M_0), \mathbf{t_0} + \varepsilon(\mathbf{t_1} - \mathbf{t_0})) = \frac{\sum_{i=1}^{k} 2(\sum_{asu} \tilde{\pi}_s P_{as} x_u (M_1 - M_0)_{iau})^2}{\tilde{P}_0^2}$$

which is continuous in $(M_1, \mathbf{t_1})$ for all $(M_0, \mathbf{t_0})$ in a neighborhood of $(W, \pi)$.

## G.5 Required Condition c): Upper bound of first derivative

With $G(S) = F(H(S), h(S))$, $\frac{\partial G((1-\varepsilon)\mathbb{D} + \varepsilon S)}{\partial \varepsilon}|_{\varepsilon=0^+} < -C < 0 \ \forall \ \pi, P$

$G(S) = f_1(H(S)) + f_2(H(S)) + f_3(H(S))$

Let $\bar{S} = ((S - \mathbb{D})t)P(\mathbb{D}t)^T + (\mathbb{D}t)P(S - \mathbb{D}t)^T$

For $f_1$

$$f_1(H((1-\varepsilon)\mathbb{D} + \varepsilon S)) = \mu_p \text{tr}(X_C^T[H((1-\varepsilon)\mathbb{D} + \varepsilon S) - \text{diag}(\sum_{j=1}^{k} H((1-\varepsilon)\mathbb{D} + \varepsilon S)_{ij})]X_C) \tag{148}$$

$$H(S) = (Sx)P(Sx)^T \tag{149}$$

$$H((1-\varepsilon)\mathbb{D} + \varepsilon S) = ((1-\varepsilon)\mathbb{D}x + \varepsilon St)P((1-\varepsilon)\mathbb{D}x + \varepsilon St)^T \tag{150}$$

$$= (\mathbb{D}x + \varepsilon(S-\mathbb{D})x)P(\mathbb{D}x + \varepsilon(S-\mathbb{D})x)^T \tag{151}$$

$$= (\mathbb{D}x)P(\mathbb{D}x)^T + \varepsilon(\mathbb{D}x)P((S-\mathbb{D})x)^T \tag{152}$$

$$+ \varepsilon((S-\mathbb{D})x)P(\mathbb{D}x)^T + \varepsilon^2((S-\mathbb{D})x)P((S-\mathbb{D})x)^T \tag{153}$$

$$\text{Finally, } f_1((1-\varepsilon)\mathbb{D} + \varepsilon S) = \text{tr}(X_C^T(\mathbb{D}x)P(\mathbb{D}x)^T X_C) + \varepsilon \text{ tr}(X_C^T(\mathbb{D}x)P((S-\mathbb{D})x)^T X_C) \tag{154}$$

$$+ \varepsilon \text{ tr}(X_C^T((S-\mathbb{D})x)P(\mathbb{D}x)^T X_C) + \varepsilon^2 \text{ tr}(X_C^T((S-\mathbb{D})x)P((S-\mathbb{D})x)^T X_C) \tag{155}$$

$$+ \text{tr}(X_C^T \text{diag}([H(\mathbb{D})_i])X_C) + \varepsilon^2 \text{ tr}(X_C^T \text{diag}([H(S-\mathbb{D})_i])X_C) \tag{156}$$

$$+ \varepsilon \text{ tr}(X_C^T \text{diag}([(((S-\mathbb{D})x)P(\mathbb{D}x)^T + (\mathbb{D}x)P(S-\mathbb{D}x)^T)_i])X_C) \tag{157}$$

$$\text{Now,} \frac{\partial f_1}{\partial \varepsilon}\bigg|_{\varepsilon=0^+} = \text{tr}(X_C^T(\bar{S} - \text{diag}([\bar{S}_i]))X_C) \tag{158}$$

For $f_2$,

$$f_2(H((1-\varepsilon)\mathbb{D} + \varepsilon S)) = \frac{\mu_p}{b}\text{tr}\left(H((1-\varepsilon)\mathbb{D} + \varepsilon S) - \text{diag}([H((1-\varepsilon)\mathbb{D} + \varepsilon S)_i])\right) - \frac{1}{b} + k - k\log b \tag{159}$$

$$\frac{\partial f_2}{\partial \varepsilon}\bigg|_{\varepsilon=0^+} = \frac{\mu_p}{b}\text{tr}\left(\bar{S} - \text{diag}([\bar{S}_i])\right) \tag{160}$$

For $f_3$,

$$f_3(H((1-\varepsilon)\mathbb{D} + \varepsilon S)) = \frac{1}{\tilde{P}_0}\text{tr}\left((\mathbb{D}x)P(\mathbb{D}x)^T + \varepsilon(\mathbb{D}x)P((S-\mathbb{D})x)^T\right. \tag{161}$$

$$\left. + \varepsilon((S-\mathbb{D})x)P(\mathbb{D}x)^T + \varepsilon^2((S-\mathbb{D})x)P((S-\mathbb{D})x)^T\right) \tag{162}$$

$$- \frac{1}{\tilde{P}_0^2}\sum_{i=1}^{k}\sum_{j=1}^{k}\left(\left[(\mathbb{D}x)P(\mathbb{D}x)^T + \varepsilon(\mathbb{D}x)P((S-\mathbb{D})x)^T\right.\right. \tag{163}$$

$$\left.\left. + \varepsilon((S-\mathbb{D})x)P(\mathbb{D}x)^T + \varepsilon^2((S-\mathbb{D})x)P((S-\mathbb{D})x)\right]_{ij}\right)^2 \tag{164}$$

$$\frac{\partial f_3}{\partial \varepsilon}\bigg|_{\varepsilon=0^+} = \frac{1}{\tilde{P}_0}\text{tr}\left(\bar{S}\right) - \frac{2}{\tilde{P}_0}\sum_{i=1}^{k}\left(\sum_{j=1}^{k}\left[(\mathbb{D}x)P(\mathbb{D}x)^T\right]_{ij} \times \sum_{j=1}^{k}[\bar{S}]_{ij}\right)^2 \tag{165}$$

From here, the proof is followed as in Appendix of Zhao et al. (2012) (in Proof of Theorem 3.1) which also borrows from Bickel & Chen (2009).

## H  Attributed SBM theory and results

We validate the robustness and sensitivity of proposed methods to variance in the node features and graph structure. We are also generating features using a multivariate mixture generative model such that the node

features of each block are sampled from normal distributions where the centers of clusters are vertices of a hypercube.

**SBM.** The Stochastic Block Model (SBM)(Nowicki & Snijders, 2001) is a generative model for graphs that incorporates probabilistic relationships between nodes based on their community assignments. In the basic SBM, a network with $p$ nodes is divided into $k$ communities or blocks denoted by $C_i$, where $i = 1, 2, \cdots, k$. The SBM defines a symmetric block probability matrix $B$ with size $(k \times k)$, where each entry $B_{ij}$ represents the probability of an edge between a node in community $C_i$ and a node in community $C_j$. Diagonal entries of this matrix represents the probabilities of intra-cluster edges. This matrix $B$ captures the intra- and inter-community connections and is assumed to be constant. $P(i \leftrightarrow j | C_i = a, C_j = b) = B_{ab}$ denotes the probability of an edge existing between nodes $i$ and $j$ when node $i$ belongs to community $a$ and node $j$ belongs to community $b$. Using these probabilities, the SBM generates a network by independently sampling the presence or absence of an edge for each pair of nodes based on their community assignments and the block probability matrix $B$.

**Degree Corrected SBM.** DC-SBM(Karrer & Newman, 2011) takes an extra set of parameters $\theta_i$ controlling the expected degree of vertex $i$. Now, the probability of an edge between two nodes (using the same notation as above) becomes $\theta_i \theta_j B_{ab}$. This was introduced to handle the heterogeneity of real-world graphs. We selected this widely used and studied model for our analysis primarily because it considers a degree parameter for all nodes, resembling a key characteristic of real-world graphs. Therefore, the DC-SBM is closer to real-world graphs than the simpler SBM while still being analyzable, making it relevant here.

**ADC-SBM Generation.** We make use of the `graph_tool` library to generate the DC-SBM adjacency matrix, with $p = 1000, k = 4$. To generate the $B$ matrix, we follow the procedure in (Tsitsulin et al., 2023), by taking expected degree for each node $d = 20$ and expected sub-degree $d_{out} = 2$. This gives us $B$ as:

$$\begin{bmatrix} 18 & 2 & 2 & 2 \\ 2 & 18 & 2 & 2 \\ 2 & 2 & 18 & 2 \\ 2 & 2 & 2 & 18 \end{bmatrix}$$

Also, $\theta$ is generated by sampling a power-law distribution with exponent $\alpha = 2$. We constrain the generated vector to $d_{min} = 2$ and $d_{max} = 4$.

To generate features, we use the `make_classification` function in the `sklearn` library. We generate a 128-dimensional feature vector for each node, with no redundant channels. These belong to $k_f$ groups, where $k_f$ might not be equal to $k$. We test three scenarios: a) matched clusters ($k_f = k$) b) nested features ($k_f > k$) c) grouped features ($k_f < k$) as visualized in Figure 5. Note that for better visualization, `class_sep` was increased to 5 (however, the results are given with a value of 1, which is a harder problem).

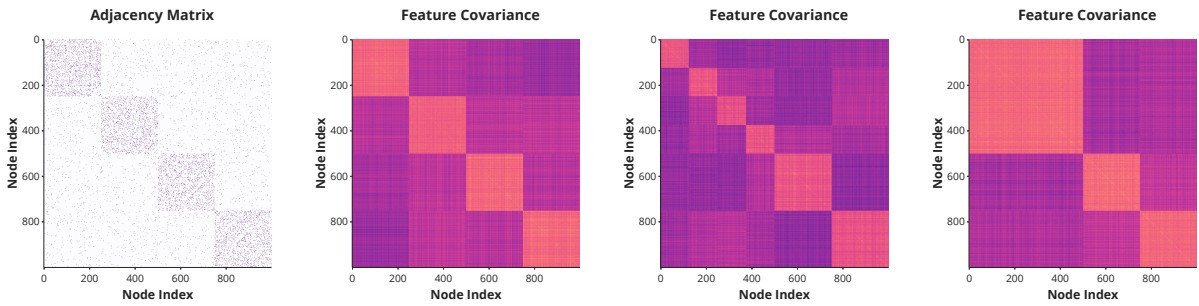

Figure (5a): Adjacency Matrix

Figure (5b): Feature Covariance Matrix $k_f = k$

Figure (5c): Feature Covariance Matrix $k_f > k$

Figure (5d): Feature Covariance Matrix $k_f < k$

Figure 5: Visualization of the generated adjacency and feature covariance matrices for the ADC-SBM

Additionally, we also consider both cases, with and without the coarsening constraint term.

**Results.** Our objective is able to completely recover the ground truth labels (NMI/ARI/ACC = 1) under all the specified conditions.

## I  VGAE

In a VGAE, the encoder learns mean ($\mu$) and variance ($\sigma$): $\mu = \text{GCN}_\mu(X, A)$ and $\log \sigma = \text{GCN}_\sigma(X, A)$ By using the reparameterization trick, we get the distribution of the latent space as: $q(Z|X, A) = \prod_{i=1}^{N} q(\mathbf{z_i}|X, A) = \prod_{i=1}^{N} \mathcal{N}(\mathbf{z_i}|\mu_i, \text{diag}(\sigma_i^2))$ A common choice for decoder is inner-product of the latent space with itself which giving us the reconstructed $\hat{A}$. $p(\hat{A}|Z) = \prod_{i=1}^{p} \prod_{j=1}^{p} p(\hat{A}_{ij}|z_i, z_j)$, with $p(\hat{A}_{ij} = 1|z_i, z_j) = \text{sigmoid}(z_i^T z_j)$

## J  Dataset Summaries and Metrics

Refer to Table 5 for the dataset summary. We use datasets directly from the `pytorch_geometric` package, so no preprocessing is needed.

**Metrics.** A pair of nodes are said to be in agreement if they belong to the same class and are assigned to the same cluster, or they belong to different classes and have been assigned different clusters. For a particular partitioning, ARI is the fraction of agreeable nodes in the graph. Accuracy is obtained by performing a maximum weight bipartite matching between clusters and labels. NMI measures the normalized similarity between the clusters and the labels, and is robust to class imbalances. Mutual Information between two labellings $X$ and $Y$ of the same data is defined as $MI(X, Y) = \sum_{i=1}^{|X|} \sum_{j=1}^{|Y|} \frac{|X_i \cap Y_i|}{N} \log \frac{N|X_i \cap Y_i|}{|X_i||Y_i|}$ and it is scaled between 0 to 1.

| Name | p ($|\mathbf{V}|$) | n ($|X_i|$) | e ($|E|$) | k ($y$) |
|---|---|---|---|---|
| Cora | 2708 | 1433 | 5278 | 7 |
| CiteSeer | 3327 | 3703 | 4614 | 6 |
| PubMed | 19717 | 500 | 44325 | 3 |
| Coauthor CS | 18333 | 6805 | 163788 | 15 |
| Coauthor Physics | 34493 | 8415 | 495924 | 5 |
| Amazon Photo | 7650 | 745 | 238162 | 8 |
| Amazon PC | 13752 | 767 | 491722 | 10 |
| ogbn-arxiv | 169343 | 128 | 1166243 | 40 |
| Brazil | 131 | 0 | 1074 | 4 |
| Europe | 399 | 0 | 5995 | 4 |
| USA | 1190 | 0 | 13599 | 4 |

Table 5: Datasets summary.

## K  Hyperparameters

### K.1  Hyperparameter Tuning

We performed hyperparameter tuning using the comet optimizer library. A mixture of grid search and Bayesian search was used for the various hyperparameters. These were the broad ranges searched for each hyperparameter:

- Learning Rate: [0.001, 0.1]
- $\alpha$: [500, 10000]
- $\beta$: [10, 250]
- $\gamma$: [100, 1000]
- $\lambda$: [0, 100]
- $\lambda_{recon}$: [10, 250]
- $\lambda_{kl}$: [0.001, 0.1]

### K.2  Sensitivity study

We also conducted a small hyperparameter sensitivity study, the resutls of which are presented in Table 6

| $\alpha$ - Coarsening Constraint | $\gamma$ - Coarsened Connectedness | $\beta$ - Modularity | NMI |
|---:|---:|---:|---|
| 5000 | 1000 | 100 | 58.6 |
| 5000 | 100 | 100 | 57.1 |
| 5000 | 100 | 10 | 57.4 |
| 1000 | 1000 | 100 | 53.5 |
| 1000 | 100 | 100 | 52.7 |
| 1000 | 100 | 10 | 52.5 |
| 10000 | 100 | 10 | 53.5 |
| 10000 | 1000 | 100 | 54.7 |

Table 6: Hyperparameter sensitivity ablation study

## L  Training Details

All experiments were run on an NVIDIA A100 GPU and Intel Xeon 2680 CPUs. We are usually running 4-16 experiments together to utilize resources (for example, in 40GB GPU memory, we can run 8 experiments on PubMed simultaneously). Again, the memory costs are more than dominated by the dataset. All experiments used the same environment running CentOS 7, Python 3.9.12, PyTorch 2.0, PyTorch Geometric 2.2.0.

## M  Results on very large datasets

The results are presented in Table 7.

| | CoauthorCS | | | CoauthorPhysics | | | AmazonPhoto | | | AmazonPC | | | ogbn-arxiv | | |
|---|---|---|---|---|---|---|---|---|---|---|---|---|---|---|---|
| Method | ACC ↑ | NMI ↑ | ARI ↑ | ACC ↑ | NMI ↑ | ARI ↑ | ACC ↑ | NMI ↑ | ARI ↑ | ACC ↑ | NMI ↑ | ARI ↑ | ACC ↑ | NMI ↑ | ARI ↑ |
| FGC | 69.6 | 70.4 | 61.5 | 69.9 | 60.9 | 49.5 | 44.9 | 38.3 | 22.5 | 46.8 | 36.2 | 23.3 | 24.1 | 8.5 | 9.1 |
| **Q-MAGC (Ours)** | 70.2 | 76.4 | 60.2 | 75.3 | 67.2 | 66.1 | 70.4 | 66.6 | **58.6** | **62.4** | 51 | 31.1 | 35.8 | 24.4 | 15.6 |
| **Q-GCN (Ours)** | 85.4 | 79.6 | 79.7 | 85.2 | **72** | **81.6** | 66.3 | 57.6 | 48.3 | 56.7 | 42.4 | 28.8 | 34.4 | 27.1 | 19.7 |
| **Q-VGAE (Ours)** | **85.6** | **79.9** | **81.6** | **86.7** | 69 | 77.7 | 69.0 | 59.4 | 49.0 | 62.3 | 45.7 | **47.2** | **39.5** | 30.4 | **24.7** |
| **Q-GMM-VGAE (Ours)** | 70.1 | 72.5 | 61.6 | 83.1 | 71.5 | 76.9 | **76.8** | **67.1** | 58.3 | 55.5 | **56.4** | 40 | OOM | OOM | OOM |
| DMoN | 68.8 | 69.1 | 57.5 | 45.4 | 56.7 | 50.3 | 61.0 | 63.3 | 55.4 | 45.4 | 49.3 | 47.0 | 25.0 | **35.6** | 12.7 |

Table 7: Comparison of methods on large attributed datasets.

## N  Evolution of different loss terms throughout training

Each separate series has been normalized by its absolute maximum value to see convergence behavior on the same graph easily. Every series is decreasing/converging (except gamma, which remains almost constant). Thus, we can be assured that no terms are counteracting and hurting the performance. The legend is provided in the graph itself. This plot Figure 6 is on the Cora dataset.

## O  Visualization of evolution of latent space

Refer to Figure 4 and Figure 7.

We can see the clusters forming in the latent space of the VGAEs. In the case of Q-VGAE, since a GCN is used on this space, it can learn non-linearities and the latent space shows different structures (like a starfish in CiteSeer). Moreoever, these structures have their geometric centres at the origin and grow out from there. In contrast, for Q-GMM-VGAE, since a GMM is being learnt over the latent space, the samples are encouraged to be normally distributed in their independent clusters, all of which have different means and comparable standard deviations. So, we see multiple "blobs", which more or less follow a normal distribution. This plot effectively shows why a GMM-VGAE is more expressive than a VGAE.

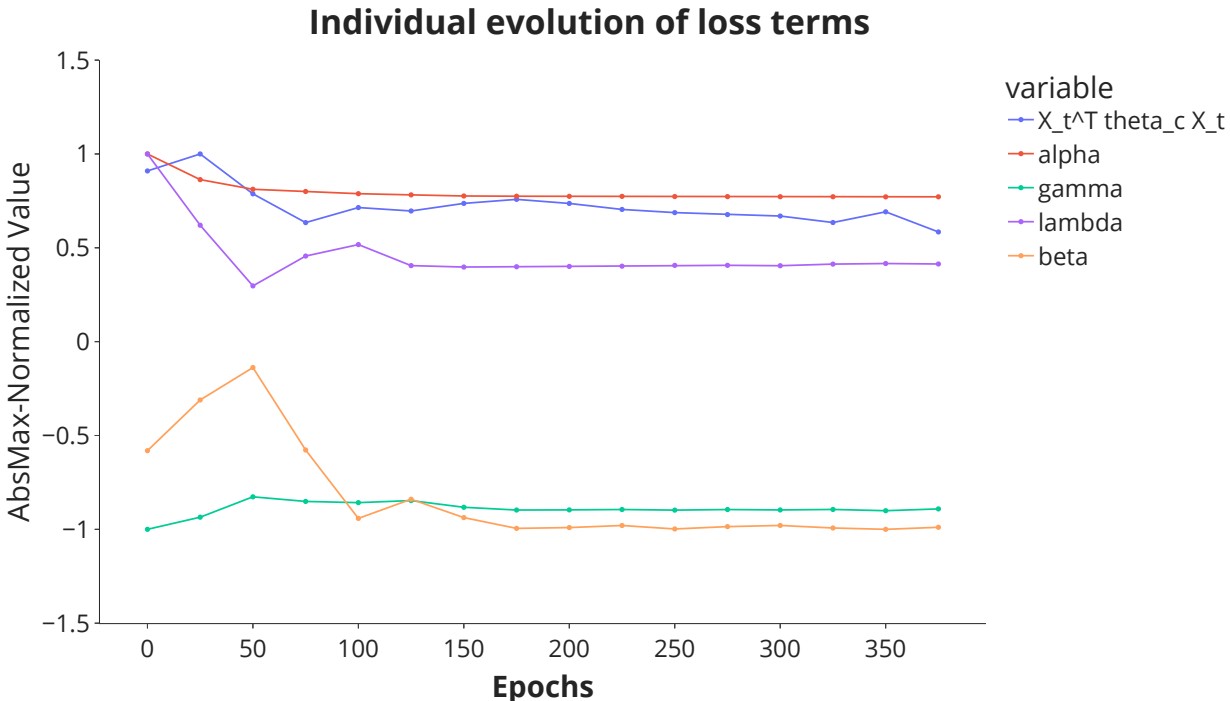

Figure 6: Evolution of the different loss terms throughout training, denoted by their weight parameters. Also the term `X_t^T theta_C X_t` term is the smoothness term $tr(X_C^T C^T \Theta C X_C)$

## P  Explanation on why VAE manifolds are curved

Embedded manifolds obtained from VGAEs are curved and must be flattened before any clustering algorithms using Euclidean distance are applied.

The latent space of a VAE is not constrained to be Euclidean. Connor et al. (2021) point out that the variational posterior is selected to be a multivariate Gaussian, and that the prior is modeled as a zero-mean isotropic normal distribution which encourages grouping of latent points around the origin. Works such as (Chen et al., 2020; Bogdanov & Shchur, 2021; Arvanitidis et al., 2018) make the VAE latent space to be Euclidean/Hyperbolic/Riemannian, and show good visuals. Moreover, it can be observed in our own work (Figure 4a and Appendix Figure 7a) that the latent manifolds are curved and so, are not suitable for conventional methods such as k-means clustering, which need Euclidean distance.

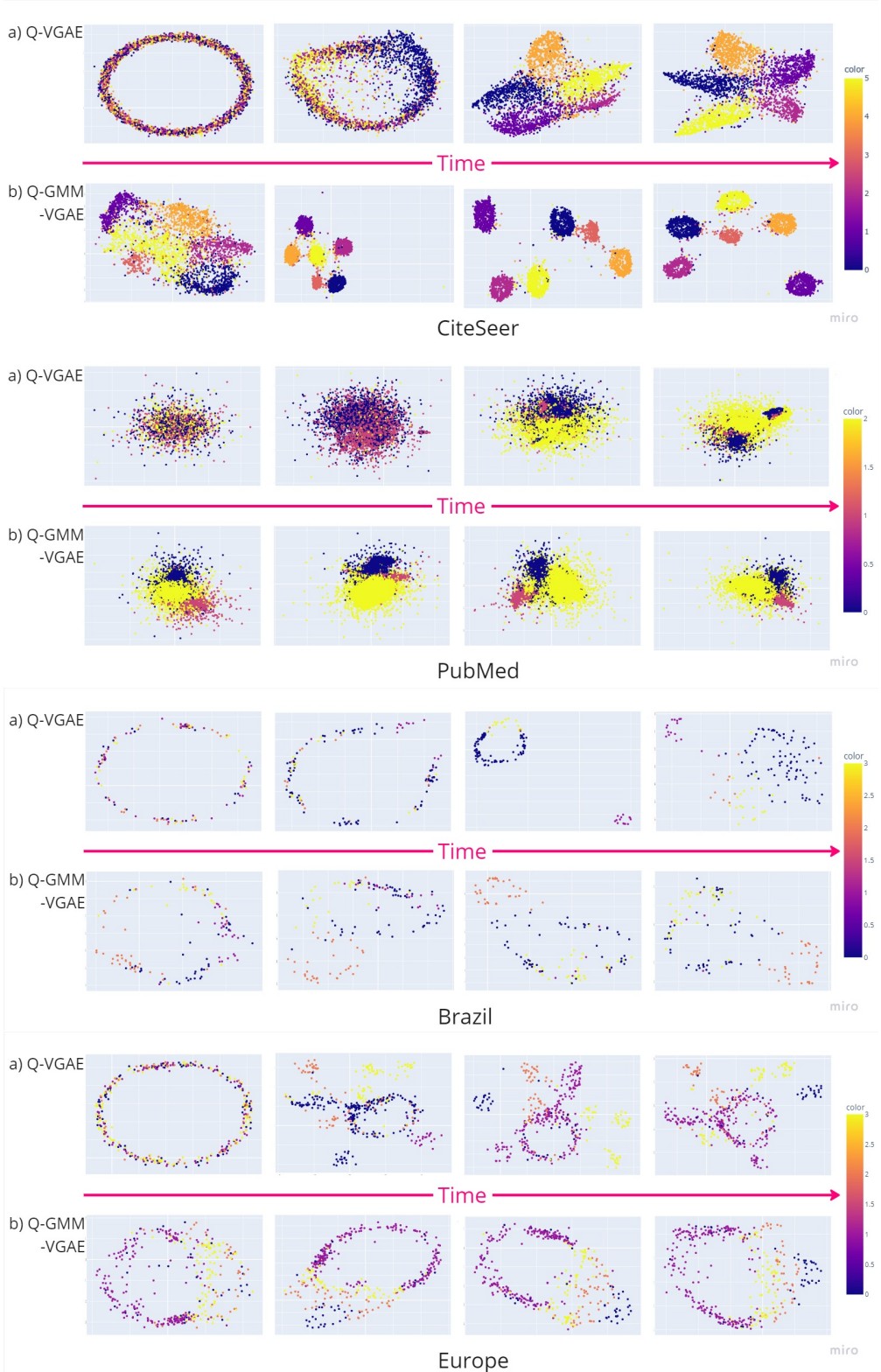

Figure 7: Plots of evolution of latent space for Q-VGAE and Q-GMM-VGAE methods for CiteSeer, PubMed, Brazil (Air Traffic) and Europe (Air Traffic) datasets.

