# OpenReview forum: "Modularity aided consistent attributed graph clustering via coarsening"
_TMLR — Accepted by TMLR_

### Review · Reviewer_BtxT · 2024-08-27

**Summary Of Contributions:**

This paper presents a novel graph clustering method via integrating coarsening and modularity maximization. The method leverages adjacency and node features,  using a loss function that consists of 3 terms, one for smoothness, one for modularity, and a log-determinant term. The authors show consistency and asymptotic error-free solutions. The authors show how to adapt the method for GNNs and VGAEs. The authors present numerical results that validate the proposed method.

**Audience:**

Yes

**Claims And Evidence:**

Yes

**Requested Changes:**

See Strengths and Weaknesses.

**Strengths And Weaknesses:**

Strengths
S1 - The paper presents a method that tackles a real-world and relevant problem.

S2 - The numerical results validate the utility of the proposed method.

S3 - The authors show theoretical results which provides evidence of why their method works.

Weaknesses
W1 - The paper is not well written. There are plenty of grammar mistakes, even in the appendix. Besides, the paper does not present the topics in order. For example, it mentions Q-MAGC before defining it. This forces the reader to go back and forth.

W2 - The concepts are not well presented. For example, in the Proposed method section, g(C,A), does not use A, but it uses B. It is unclear if it should use A, or B, and even who is B? Same with h(C,\Theta), which uses J, who is J? The authors should be more careful about the math presentation.

W3 - There seems to be a mismatch between theory and experiments. The theory works under certain assumptions over the matrices, which do not hold once the GNNs and VGAEs are introduced. Besides, the authors say things such as “using a “soft” and “hard” version of C”. The authors should mathematically define what they mean by this.

W4 - The contribution of the paper seems to be the problem formulation. However, this is nothing but an addition to existing loss functions. That is to say, the authors simply put three loss functions together, and claim things about it. This is problematic in my opinion for several reasons. In terms of novelty, the author's contribution is minor. Even more so, the authors simply put the three pieces together, without even showing why this makes sense from a magnitude standpoint. That is to say, why of the three terms govern the minimization? And by how much? The authors never show the loss per term in the appendix. When looking at Figure 7, it is clear that all losses converge very fast and to different values. I suggest the authors explore a better way of combining the terms.

W5 - The authors should elaborate more on the DC-SBM definition. It seems to be a very simple model for graphs, can the authors explain why such a simplistic model is relevant in practice? Also, can the authors provide synthetic experiments showcasing their method on it?

---

> ### Author Response · Authors · 2024-10-05
> **Response by authors (part 1 of 4)**
>
> We thank the reviewer for the encouraging feedback, which has greatly helped us in improving our paper.
>
> ### Strengths
> S1 - The paper presents a method that tackles a real-world and relevant problem.
>
> S2 - The numerical results validate the utility of the proposed method.
>
> S3 - The authors show theoretical results which provides evidence of why their method works.
>
> Your recognition of these strengths of our study, particularly the comment that our exhaustive numerical results are backed by theoretical evidence is highly encouraging.
>
>
> ### Weaknesses / Requested Changes
>
> ### W1
> The paper is not well written. There are plenty of grammar mistakes, even in the appendix. Besides, the paper does not present the topics in order. For example, it mentions Q-MAGC before defining it. This forces the reader to go back and forth.
>
> **A1** - We thank the reviewer for pointing this out. We have thouroughly checked the paper for grammar mistakes and addressed them in the latest revision, which were an inadvertent oversight. Additionally, we have clarified the initial reference to Q-MAGC to properly introduce the method's name. Please let us know if any further clarifications are needed.
>
> ---
>
> ### W2
> The concepts are not well presented. For example, in the Proposed method section, g(C,A), does not use A, but it uses B. It is unclear if it should use A, or B, and even who is B? Same with h(C,\\Theta), which uses J, who is J? The authors should be more careful about the math presentation.
>
> **A2** - $B$ is the modularity matrix, as defined in Equation 3 on Page 4 of the background section, is fundamental to the paper. Moreover, $g(C, A)$ uses $B = A - \frac{dd^T}{2e}$, where $d = A.\mathbb{1}$, is a function of $A$. We have now written this out more clearly.
> Regarding $J$, we define it within the same equation where it is introduced (eqn 5), quoting
> > $$\text{where, } J = \frac{1}{k}\mathbb{1}_{k\times k}$$.
>
> We hope this explanation addresses your concern. We would be happy to answer any further questions.
>
> ---

---

> ### Author Response · Authors · 2024-10-05
> **Response by authors (part 2 of 4)**
>
> ### W3
> There seems to be a mismatch between theory and experiments. The theory works under certain assumptions over the matrices, which do not hold once the GNNs and VGAEs are introduced. Besides, the authors say things such as “using a “soft” and “hard” version of C”. The authors should mathematically define what they mean by this.
>
> **A3** - We have done exhastive experimentation for both cases where the theory directly holds and where it indirectly holds, shown in section 6. The results of the `Q-MAGC` are the ones pertaining directly to the discussed theory (i.e. GNN-free). This works well and surpasses all traditional non-deep learning methods and many deep-learning methods as well (Table 1).
>
> We further explore methods to enhance `Q-MAGC`'s performance by using GNNs/VGAEs (`Q-GCN`/`Q-VGAE`) which are known for learning improved graph representations through message passing. It is important to note that these models are significantly more complex and challenging to analyze theoretically.
>
> Now, coming to our next theoretical result on consistency, the goal is to determine whether the optimization objective or loss can accurately recover the correct labeling (cluster assignment). To evaluate the performance of our loss function, we need to make certain assumptions about the underlying graph. The most standard approach is to leverage the stochastic block model (SBM) framework, a generative model for random graphs, which is commonly used for benchmarking algorithms that recover community structures in graph data. We selected the Degree-Corrected SBM (DC-SBM) for our analysis primarily because it considers a degree parameter for all nodes, resembling a key characteristic of real-world graphs. DC-SBM is a good choice as it not too complex to analyze while also not being too far from real world graphs. Proving that our loss is strongly and weakly consistent under the DC-SBM makes our analysis tighter and shows that our method is robust. We also want to add that this result holds for loss function and on the optimization algorithm.
>
> Nevertheless, given the strong theoretical properties of `Q-MAGC`, we were confident that integrating it with GNNs would likely improve performance, which is validated by our empirical results. Interestingly, our loss function can be integrated with any GNN-based architecture and it would lead to performance improvements for clustering. We demonstrate this with the complex GMM-VGAE architecture as well (`Q-GMM-VGAE`).
>
> Previous literature also follows suit here: most analyses such as in [Tsitsulin et al. 2023](https://jmlr.org/papers/v24/20-998.html) have been done on a traditional block model rather than on GCNs/ other deep networks, because of the above mentioned difficulties.
>
> We hope this explanation helped and we would be happy to answer further questions about our theoretical analyses.
>
> **On 'soft' and 'hard' terminology**
> By *soft* vs. *hard*, we wanted to imply the usual ML meaning of probability based vs. label based. That is, *soft* $C$ means $C_{i,j}$ is the probability that node $i$ belongs to cluster $j$ ($C_{i,j} \in [0, 1]$), and *hard* $C$ means $C_{i,j}$ is the binary assignment ($C_{i,j} \in \{0,1\}$) whether node $i$ belongs to cluster $j$ or not. The hard assignment is found using a per-row `argmax` over the soft assignment (i.e. selecting the highest probability cluster for each node).
> We have now incorporated this into the paper.
>
> ---

---

> > ### Author Response · Authors · 2024-10-05
> > **Response by authors (part 3 of 4)**
> >
> > ### W4
> > The contribution of the paper seems to be the problem formulation. However, this is nothing but an addition to existing loss functions. That is to say, the authors simply put three loss functions together, and claim things about it. This is problematic in my opinion for several reasons. In terms of novelty, the author's contribution is minor. Even more so, the authors simply put the three pieces together, without even showing why this makes sense from a magnitude standpoint. That is to say, why of the three terms govern the minimization? And by how much? The authors never show the loss per term in the appendix. When looking at Figure 7, it is clear that all losses converge very fast and to different values. I suggest the authors explore a better way of combining the terms.
> >
> > **A4** - We respectfully disagree with the reviewer. To summarise, our contribution is two-fold:
> >
> > First, we design a robust optimization objective that can be solved independently, and has several important theoretical properties such as convergence, low time complexity, and is consistent. The chosen terms complement each other and can only learn a cluster assignment matrix with the desired properties when used together. Choosing these terms from a wide gamut of possibilities was not straightforward, and required consideration about the properties and behaviour of each term.
> >
> > Modularity and coarsening in general have been studied before but their combination has not been explored. Any consisitency analysis of such a function has also not been performed before.
> >
> > Second, we show the seamless integration of our objective with any GNN-based architecture for enhanced performance, by taking advantage of the message passing in GNNs.
> >
> > The tight convergence, optimality, and consistency analysis are unique to our paper and are often overlooked in other clustering-based methods that use either modularity maximization or deep learning.
> >
> > Usually in previous literature assumptions are taken in order to show some theoretical properties which dont hold often in real world settings. For example, in [Tsitsulin et al. 2023](https://jmlr.org/papers/v24/20-998.html), the assumption taken in order to prove consistency is that the graph has a positive modularity under the ground truth labels. However, that is not true for 3 popular datasets in our own testing: Brazil, USA and Europe (Airports datasets). On the other hand, we have verified that the approximation that we take will hold on real world graphs (Appendix D, equation 93 onwards).
> >
> > We list our contributions under the **Key Contributions** subheading in the Introduction section (page 2). We had to extensively research and select terms so that the resulting loss function would be both theoretically sound and practically useful, while also making it able to be optimized by gradient descent in neural networks. We have also explained these in the Methods section.
> >
> > In Figure 3.b), we demonstrate the performance improvement when each term is added to the objective. We discuss these findings in the ablation studies section of the paper.
> >
> > **Regarding individual loss terms**
> >
> > Quoting from the paper:
> > > We analyze the evolution of the different loss terms during training ... as shown in Appendix N.
> >
> > We have shown in detail the behaviour of our loss terms (and the evolution of each term) in Appendix N, and figure 7 (on page 33-34).
> > Here, it is important to note that since the losses correspond to different terms they will converge at different rates and to different values. Also, the total loss will not go to 0, but will be minimized overall (because some terms such as the trace term for smoothness can never be 0). Following that, we can see in Fig 7 of Appendix N that all terms coverge to some minimum value.
> >
> > Moreover, we go a step further and show the sensitivity of each term's coefficient to the performance in the ablation studies. We hope this answer resolved your concerns and we would be happy to discuss more and clear any remaining issues.
> >
> > ---

---

> > > ### Author Response · Authors · 2024-10-05
> > > **Response by authors (part 4 of 4)**
> > >
> > > ### W5
> > > The authors should elaborate more on the DC-SBM definition. It seems to be a very simple model for graphs, can the authors explain why such a simplistic model is relevant in practice? Also, can the authors provide synthetic experiments showcasing their method on it?
> > >
> > > **A5** - To show that our loss function is consistent, we need to make certain assumptions about the underlying graph. The most standard approach is to use the stochastic block model (SBM) framework, a generative model for random graphs. We selected the widely used and studied DC-SBM model (modified version of SBM) for our analysis primarily because it considers a degree parameter for all nodes, resembling a key characteristic of real-world graphs. Therefore, the DC-SBM is closer to real-world graphs than the simpler SBM while still being analyzable, making it relevant for our paper. Some details about it are given in Appendix I (pages 30 and 32). Here we also mention that
> > > > This (DC-SBM) was introduced to handle the heterogeneity of real-world graphs.
> > >
> > > [Karrer and Newman (2011)](https://doi.org/10.1103%2FPhysRevE.83.016107) also explain why DC-SBMs are better:
> > > > Existing SBMs usually ignore the variation in the node degrees in real-world networks. Under the original SBM, the expected degree is the same for all nodes in each group.
> > >
> > > Analyzing the modularity term using SBMs won't yield meaningful results, as SBMs assume same expected degree for all nodes. This is important because the modularity matrix is a function of the degree vector of a graph, making DC-SBM an optimal choice for our optimization formulation.
> > >
> > > As for the experiments, you can also find them in Appendix I under the Results subheading (page 32). For convenience:
> > > > Our objective is able to completely recover the ground truth labels (NMI/ARI/ACC = 1) under all the specified conditions.
> > >
> > > We hope this explanation helped.
> > >
> > > Some other examples of DC-SBMs being used in literature:
> > > - Newman, M. and Reinert, G. (2016) 'Estimating the Number of Communities in a Network', _Physical Review Letters_, 117(7). Available at: https://doi.org/10.1103/physrevlett.117.078301.
> > > - Zhao, Y., Levina, E. and Zhu, J. (2012) 'Consistency of community detection in networks under degree-corrected stochastic block models', _The Annals of Statistics_, 40(4). Available at: https://doi.org/10.1214/12-aos1036.
> > > - Wan, Y. and Meila, M., 2016. Benchmarking recovery theorems for the DC-SBM. In _ISAIM_. Available at: https://sites.stat.washington.edu/people/mmp/Papers/isaim16-comparison.pdf.
> > > - Avrachenkov, K. and Dreveton, M. (2020) 'Almost exact recovery in noisy semi-supervised learning', _arXiv (Cornell University)_ \[Preprint\]. Available at: https://doi.org/10.48550/arxiv.2007.14717.
> > > - Tsitsulin, A. _et al._ (2023) 'Graph Clustering with Graph Neural Networks', _Journal of Machine Learning Research_, 24(127), pp. 1–21. Available at: https://jmlr.org/papers/v24/20-998.html.
> > >
> > > We hope this response addresses your concern.
> > >
> > > ---
> > >
> > > Thank you once again for your valuable insights and support.

---

> > > > ### Author Response · Authors · 2024-10-05
> > > > **We have added a revision**
> > > >
> > > > We have added a revision to the paper incorporating your feedback.
> > > > - Clarified the first reference of Q-MAGC.
> > > > - Defined the soft and hard versions of the cluster assignment matrix C more clearly.
> > > > - Corrected grammatical errors.
> > > > - Added the reason behind using DC-SBM in Appendix I.
> > > >
> > > > Here is the [PDF diff (anonymized link)](https://anonymous.4open.science/api/repo/MAGC-8880/file/diffchecker%20rebuttal.pdf?v=afcb6801) for convenience.\
> > > > On the left side is the old version, and on the right side is the new version.\
> > > > Text in red is removed, in green is added and in blue is just moved without edits.
> > > >
> > > > We would be happy to answer any questions that you may have.

---

> > > > > ### Author Response · Authors · 2025-03-01
> > > > > **Summary of changes in the latest revision and PDF diff**
> > > > >
> > > > > Dear reviewer, we had already mentioned your changes in the comments replying to your review. We hope you were satisfied with our response. \
> > > > > Here is the latest list of changes based on feedback from the other reviewers. We will be happy to answer any questions you have.
> > > > >
> > > > > # Changes
> > > > > Based on the feedback received from the reviewers, we have made the following changes. We have uploaded a PDF with the changes colored (as the main submission PDF). A side-by-side diff PDF of the changes highlighted from the last version is available at https://anonymous.4open.science/api/repo/MAGC-8880/file/TMLR%20Final%20Diff.pdf
> > > > >
> > > > > ## Major Changes
> > > > > - Merged Background and Related Works sections to make it concise and improve readability.
> > > > > - Moved  the detailed literature survey to Appendix B.
> > > > > - Explain why log-det term helps with inter-connectivity.
> > > > > - Added small proof for deriving the majorized function (Equations 7-10).
> > > > > - Added a proof sketch for the convergence analysis.
> > > > > - Added reasons for choosing DC-SBM for the consistency analysis in Section 3.
> > > > > - Moved the consistency proof to Appendix G and added a proof sketch in the main paper.
> > > > > - Clarified the importance of the consistency analysis in the Introduction and Section 3.
> > > > > - Added a datasets table in the main paper (Table 1).
> > > > > - Added hyperparameter tuning details in Appendix K.1
> > > > > - Moved figure showing evolution of various loss terms with training from the appendix to the main paper (Figure 3b).
> > > > > - Added a hyperparameter sensitivity ablation study in Appendix K.2
> > > > > - Added a discussion on the visualization of latent spaces.
> > > > > - Added a discussion on the performance of Q-GCN/VGAE/GMM-VGAE.
> > > > > - Reordered the appendices based on first occurence.
> > > > >
> > > > > ## Minor Changes
> > > > > - Improved the captions of the figures and increased sizes.
> > > > > - Clarified meaning of Q-MAGC in the Introduction.
> > > > > - Fixed minor grammatical and typing errors in the paper.

---

> > > > > > ### Comment · Reviewer_BtxT · 2025-04-02
> > > > > > **Acknowledgment of Reply**
> > > > > >
> > > > > > The changes I proposed have been addressed.
> > > > > > Thank you.

---

### Review · Reviewer_33er · 2024-12-20

**Summary Of Contributions:**

This paper proposed a graph clustering method which combined graph coarsening and spectral modularity maximization. The authors showed the theoretical background of the proposed method including KKT optimality, convergence analysis, and consistency analysis on degree corrected stochastic block models (DC-SBM). The proposed method could be integrated with graph neural network (GNN), graph convolutional network (GCN), variational graph auto-encoders (VGAE), and Gaussian Mixture Model VGAE (GMM-VGAE). In the experimental results, the proposed method was applied to real-world datasets such as Cora, CiteSeer, PubMed, and Airports (Brazil, Europe, and USA) for attributed or non-attributed graph clustering.

**Audience:**

Yes

**Broader Impact Concerns:**

There is no related concern.

**Claims And Evidence:**

No

**Requested Changes:**

1) The authors wrote that both graph coarsening and modularity maximization lacked theoretical convergence guarantees in p. 4. The proposed method seems to simply combine graph coarsening method and modularity maximization method - how did it achieve convergence?
2) What improvements can we expect from being consistent on DC-SBM?
3) It might be better to omit the section 3 background. The undefined terminologies in equations (1) and (2) adds to the confusion, and it would be easier to understand if equation (4) is explained alongside equation (5).
4) The consistency analysis on DC-SBM in the section 4 takes up too much space in the paper. It would be better to move this part to the appendix and mention the meaning and advantages of consistency analysis.
5) It would be helpful to include the following items in the results section.
- The explanations for the figures and tables were lacking.
- It would be better to provide more details about the dataset in the results section.
- Where can we find the results for the very large datasets, CoauthorCS/Physics, AmazonPhoto/PC, and ogbn-arxiv?
- The authors proposed 4 new methods, Q-MAGC, Q-GCN, Q-VGAE, and Q-GMM-VGAE in Table 1. So, it is difficult to find which one is superior. It would be better to demonstrate the results and comparisons to indicate when and how to use the 4 different proposed methods in the results section.
- Figures 2 and 3 are too small, making it difficult to read the text.
- The most important aspect of this paper seems to be demonstrating that combining the graph coarsening term and the modularity maximization term in equation (5) improved performance without significantly affecting the running time. However, this is only briefly mentioned in the results section and refers the reader to the appendix. Additionally, it's unclear what the figures in the appendix are showing.
- Figure 4 showed the latent spaces of Q-VGAE and Q-GMM-VGAE. What did they mean?

**Strengths And Weaknesses:**

This paper proposed new graph clustering methods to combine graph coarsening and spectral modularity maximization, showed the theoretical background of the proposed method. Furthermore, the authors demonstrated the potential to improve performance by integrating their proposed method with various deep learning-based approaches.
However, there is no discussion on what the theoretical background means and how it can be utilized. Additionally, it might be better to reorganize the overall structure, and it seems that the results section does not adequately show the superiority of the proposed method.

---

> ### Author Response · Authors · 2025-03-01
> **Response by authors (part 1 of 2)**
>
> We thank the reviewer for the constructive feedback. We're grateful for the recognition of our paper's strengths, particularly the theoretical guarantees along with the empirical evaluations. Additionally, we appreciate your encouraging comments which have greatly contributed to improving the quality of our work. Thank you for your thoughtful suggestions!
>
> Addressing the weaknesses raised by the reviewer, we have rewritten significant portions of the paper to retain only the relevant theoretical background in the main paper and moved the proofs to the appendix. Additionally, we have restructured the paper to improve readability. We have also elaborated on the importance of the consistency proof in both the main paper and this rebuttal to further highlight its relevance.
>
> ## Requested Changes:
>
> ### Q1
> **The authors wrote that both graph coarsening and modularity maximization lacked theoretical convergence guarantees in p. 4. The proposed method seems to simply combine graph coarsening method and modularity maximization method - how did it achieve convergence?**
>
> **A1** -
>
> While modularity maximization may be relatively standard, graph coarsening doesn't have an mutually agreed-upon framework, with techniques varying from MinCut based methods, it's many variants, to more exotic methods such as Local Variation [Loukas, 2019](https://jmlr.org/papers/v20/18-680.html).
> The chosen terms complement each other and can only learn a cluster assignment matrix with the desired properties when used together. Choosing these terms from a wide gamut of possibilities was not straightforward, and required consideration about the properties and behaviour of each term.
>
> This makes our method a **delicate balance** between maximizing modularity and coarsening the graph. Each term is chosen carefully while taking their interaction into account - so that no two terms work against each other\*.
>
> This can also be verified from the loss over time plots in appendix N, where plot each term separately.
>
> \* - *Note that this can only be managed to an extent, since the highest modularity solution usually does not correspond to the ground truth clustering, but gets us close, as mentioned in Section 6.5, p 12.*
>
> ### Q2
> **What improvements can we expect from being consistent on DC-SBM?**
>
> **A2** -
> To show that our loss function is consistent, we need to make certain assumptions about the underlying graph. The most standard approach is to use the stochastic block model (SBM) framework, a generative model for random graphs. We selected the widely used and studied DC-SBM model (modified version of SBM) for our analysis primarily because it considers a degree parameter for all nodes, resembling a key characteristic of real-world graphs. Therefore, the DC-SBM is closer to real-world graphs than the simpler SBM while still being analyzable, making it relevant for our paper.
> We have added the above to the main paper, under the Consistency Analysis heading in Experiments Section 5.
>
> Some details about it are given in Appendix . Here we also mention that
> > This (DC-SBM) was introduced to handle the heterogeneity of real-world graphs.
>
>
> Real world graphs can be seen as the combination/superposition of different models such as multiple DC-SBMs, since one model is usually too scope-limited for the wide range of phenomenon where graphs are a natural way of representing things. Our method being strongly consistent on DC-SBM (i.e. cluster assignment error rate tends to 0) implies that it will work provably well on parts of the graph which follows this model, and we can reasonably expect those benefits to spread to other related graph structures - either directly (provably) or indirectly (*similarity*).
>
> ### Q3
> **It might be better to omit the section 3 background. The undefined terminologies in equations (1) and (2) adds to the confusion, and it would be easier to understand if equation (4) is explained alongside equation (5).**
>
> **A3** -
> We thank the reviewer for this suggestion. We have merged the Related Works and Background sections to make it more concise. This section introduces key concepts related to coarsening and modularity maximization which are not discussed anywhere else in the paper and summarizes previous related papers. We have removed extra terms defined in this section which are not used in the main paper and moved them to the appendix. The detailed literature survey can be found in Appendix B.
>
> Please let us know if you feel more changes are needed to improve this section.

---

> ### Author Response · Authors · 2025-03-01
> **Response by authors (part 2 of 2)**
>
> ### Q4
> **The consistency analysis on DC-SBM in the section 4 takes up too much space in the paper. It would be better to move this part to the appendix and mention the meaning and advantages of consistency analysis.**
>
> **A4** - We thank the reviewer for this suggestion. Since it is a core contribution of the paper, we haven't completely removed consistency analysis from the main paper, but instead have given a smaller proof sketch in the main paper and shifted the detailed proof to Appendix D. Additionally, we have dicussed the advantages offered by a strongly consistent formulation and the reasons for choosing the DC-SBM framework in Section 3.
>
> ### Q5
> **It would be helpful to include the following items in the results section.**
>
> 1. **The explanations for the figures and tables were lacking.**\
> **A 5.1** - We have improved the caption and in-text explanation of Figures 2 and 3b. As for the other short-caption (Fig 4), it is shown alongside the paragraph that describes it in detail. We have also adjusted the positions of tables and figures and organised the paper better. We thank the reviewer for this suggestion.
>
> 2. **It would be better to provide more details about the dataset in the results section.**\
> **A 5.2** - We thank the reviewer for this suggestion, we have move details about datasets to the Experiments Section 5.
>
> 3. **Where can we find the results for the very large datasets, CoauthorCS/Physics, AmazonPhoto/PC, and ogbn-arxiv?**\
> **A 5.3** - The results for these large datasets are given in Appendix M, table 7.
>
> 4. **The authors proposed 4 new methods, Q-MAGC, Q-GCN, Q-VGAE, and Q-GMM-VGAE in Table 1. So, it is difficult to find which one is superior. It would be better to demonstrate the results and comparisons to indicate when and how to use the 4 different proposed methods in the results section.**\
> **A 5.4** - Q-MAGC (without GNNs) is well suited for light/edge applications where efficiency and reproducibility is key, since it is deterministic.
> For other applications, the order is Q-GCN < Q-VGAE\Q-GMM-VGAE.\
> \
> The Q-GCN architecture assigns the role of both embedding the graph and predicting the assignment matrix to the same network, so it is understandably less performant.\
> \
> In most cases, Q-VGAE should serve better, considering the extra GMM pretraining that is required for the GMM-VGAE architecture.
> Depending on underlying data distribution, Q-GMM-VGAE might work better if the distribution is very complex and would help from separate priors being learnt for each cluster.\
> \
> We have added this in the Discussion section (Section 6).
>
> 5. **Figures 2 and 3 are too small, making it difficult to read the text.**\
> **A 5.5** - We thank the reviewer for this suggestion and we have corrected this in the latest revision.
>
> 6. **The most important aspect of this paper seems to be demonstrating that combining the graph coarsening term and the modularity maximization term in equation (5) improved performance without significantly affecting the running time. However, this is only briefly mentioned in the results section and refers the reader to the appendix. Additionally, it's unclear what the figures in the appendix are showing.**\
> **A 5.6** - We discuss running time in the Experiments Section 5.5 which includes Figure 2 and a discussion on the running times of different methods. From Figure 2, it is clear that our Q-variants of methods - GCN, VGAE are better than vanilla. We also discuss the complexity of Algorithm 1 and compare it to other graph custering methods in Appendix F - this section does not include any figures. We would be happy to answer further questions.
>
> 7. **Figure 4 showed the latent spaces of Q-VGAE and Q-GMM-VGAE. What did they mean?**\
> **A 5.7** -
> Figure 4 shows that the embeddings for both Q-VGAE and Q-GMM-VGAE improve as training progresses.
> Q-VGAE starts from random initialization whereas Q-GMM-VGAE is initialized with the weights learnt from GMM-VGAE.
> It shows that Q-GMM-VGAE which uses a pre-trained GMM-VGAE model learns a better representation i.e. clusterable embeddings which improve over time.
> We have added this discussion in Section 5.5

---

> > ### Author Response · Authors · 2025-03-01
> > **Summary of changes in the latest revision and PDF diff**
> >
> > # Changes
> > Based on the feedback received from the reviewers, we have made the following changes. We have uploaded a PDF with the changes colored (as the main submission PDF). A side-by-side diff PDF of the changes highlighted from the last version is available at https://anonymous.4open.science/api/repo/MAGC-8880/file/TMLR%20Final%20Diff.pdf
> >
> > ## Major Changes
> > - Merged Background and Related Works sections to make it concise and improve readability.
> > - Moved  the detailed literature survey to Appendix B.
> > - Explain why log-det term helps with inter-connectivity.
> > - Added small proof for deriving the majorized function (Equations 7-10).
> > - Added a proof sketch for the convergence analysis.
> > - Added reasons for choosing DC-SBM for the consistency analysis in Section 3.
> > - Moved the consistency proof to Appendix G and added a proof sketch in the main paper.
> > - Clarified the importance of the consistency analysis in the Introduction and Section 3.
> > - Added a datasets table in the main paper (Table 1).
> > - Added hyperparameter tuning details in Appendix K.1
> > - Moved figure showing evolution of various loss terms with training from the appendix to the main paper (Figure 3b).
> > - Added a hyperparameter sensitivity ablation study in Appendix K.2
> > - Added a discussion on the visualization of latent spaces.
> > - Added a discussion on the performance of Q-GCN/VGAE/GMM-VGAE.
> > - Reordered the appendices based on first occurence.
> >
> > ## Minor Changes
> > - Improved the captions of the figures and increased sizes.
> > - Clarified meaning of Q-MAGC in the Introduction.
> > - Fixed minor grammatical and typing errors in the paper.

---

### Review · Reviewer_mKbj · 2025-02-18

**Summary Of Contributions:**

- This paper presents an optimization-based framework for attributed graph clustering that integrates graph coarsening with modularity maximization. Traditional methods, such as cut-based, similarity-based, and modularity-based approaches, often face challenges in accurately capturing community structures, handling computational complexity, and detecting smaller communities.

- To address these limitations, this work introduces the Q-MAGC loss function, which combines modularity maximization, graph smoothness, and log-determinant regularization. This formulation enhances clustering performance and guarantees asymptotically error-free clustering under the Degree-Corrected Stochastic Block Model (DC-SBM).

- The proposed approach is modeled as a multi-block non-convex optimization problem and is solved using the Block Majorization-Minimization technique.

- The method is implemented in GNNs and VGAEs, introducing three models—Q-GCN, Q-VGAE, and Q-GMM-VGAE—which demonstrate state-of-the-art performance on benchmark datasets.

**Audience:**

Yes

**Broader Impact Concerns:**

There are no significant ethical concerns in this work, as it focuses on methodological advancements in graph clustering without involving sensitive data, privacy issues, or societal harm

**Claims And Evidence:**

Yes

**Requested Changes:**

[Limitations requiring Revision]

<Major Concerns>

(1) The paper reads more like a Ph.D. thesis rather than a journal article. The attempt to cover too many aspects makes it difficult to read smoothly, and it is challenging to follow through to the end. A better approach might be to feature the core findings in this paper and consider splitting other aspects, such as convergence analysis and consistency analysis, into a sequel paper.

(2) The proposed optimization framework is quite similar to Kumar's (2023, JMLR), except for the addition of the modularity term \( tr(C^TBC) \) in the objective function—Eq. (5). By incorporating this term, the problem is no longer convex, leading the authors to propose an optimization approach using MM. Therefore, the distinction from Featured Graph Coarsening (FGC) in Kumar’s work should be clearly highlighted. Additionally, comparative studies should be conducted more thoroughly, rather than treating FGC as merely one of the comparative methods in the current experimental setup.

(3) Particularly, the ablation study for each term in the loss function needs to be emphasized more clearly, rather than being placed in the Appendix as in the current paper. Additionally, the legends in Figure 7 are difficult to recognize. Please improve their readability and check the font size of other figures to ensure clarity.

(4) The related works presented in the Introduction and Section 2: Related Works are somewhat confusing. Narrow the scope, present them in a more integrated manner, and explicitly explain their relevance to the proposed method.

(5) In the paragraph below Eq.(4), the coarsening objective \( f \) is described as preserving inter-connectivity, while in Eq.(5), \( h \) is stated to maintain inter-connectivity. This is confusing. Simply put, in the coarsened graph, \( f \) should be responsible for maintaining the smoothness of node features, while \( h \) should preserve inter-connectivity. However, why does increasing \( -\log \det (C^\top \Theta C + J) \) lead to such effects? Please provide a more detailed explanation.

(6) To show the \(f(C)\) in Eq.(6) is `L-Lipschitz continuous gradient function', each term of which is proved as a `L-Lipschitz continuous' in Appendix B. However, Lipschitz continuity does not necessarily means Lipschitz-Continuous Gradient of \(f(C)\). So the more explanation is needed.

(7) It is difficult to follow the transition to Eq(8). How is Eq.(8) formulated as a majorized problem of Eq.(5)? Since one of the main contributions of the proposed method is the majorization of the objective function, a more detailed explanation of the procedure should be provided.

(8) In Appendix B, Equation (62) defines the optimal solution satisfying the KKT conditions as:
            \[ C = \frac{(A)^+}{\sum_i ||[A^T]_i||_2} \].
            However, \( A \), as defined below Equation (55), is given by:
            \[ A = \left( C - \frac{1}{L} \nabla f(C) \right)^+\]
            which is a recursive formulation. How can this be solved?

(9) Moving the Lagrangian equation (Eq. 55) from Appendix B to the main manuscript will enhance the readability of the proposed method.

(10) The section on consistency analysis with DC-SBM takes up too much space. It would be better to shorten this part, similar to how convergence analysis and complexity analysis are presented.

(11) Hyperparameter tuning and data preprocessing details are not sufficiently discussed, making replication and comparison more difficult.

(12) Important figures and tables for key experiments are placed in the appendix. Switching the proofs to Appendix and moving the experimental setup and results to the main text would provide better accessibility for a wider range of readers.

(13) The authors acknowledge that the proposed approach relies heavily on modularity.  In datasets where modularity is naturally low (e.g., the Airports dataset), the performance advantage over other methods diminishes.  This raises concerns about the generalizability of the method to graphs with weakly modular structures.

(14)  Please revise the reference list to follow a standard format, either chronological or alphabetical, to improve clarity. Currently, it is difficult to locate specific citations within the list.

(15) Magnify the font size of markers and legends in Figure 2 for better readability. Additionally, Figure 2(a) is presented as a figure, but it is actually a table.  The misuse of object types due to space constraints should be avoided. A similar issue is found in Figure 3, which should also be corrected. Ensure that figures and tables are appropriately categorized to maintain clarity and accuracy in presentation.

(16) Align the appendix sections to match their order of appearance in the main manuscript.


<Minors>

(17) On page 2, when Q-MAGC first appears, write out its full name and provide appropriate references.

(18) On page 3, provide appropriate references for block maorization-minimization (Block MM) techniques.

(19) Provide the full name for DC-SBM when it first appears.

(20) On page 5, when Q-MAGC first appears, write out its full name to ensure clarity for readers encountering it for the first time

(21) Eq. (5),(6)  Typo: \(C \in R^{p \times l}\) \( l \) is not correct.

(22) In Appendix A, Equation (29), add a `+' sign before the last term.

(23) In Appendix A, Equation (31), replace \( V \) with \( U \).

(24) It is recommended that the Lipschitz constants \( L_1, L_2, L_3, L_4 \) be assigned in the order in which they appear in either equation (5) or (6).

**Strengths And Weaknesses:**

[Strengths and Contributions]
- The paper provides theoretical soundness and convergence guarantee of the proposed optimization framework: Karush-Kuhn-Tucker(KKT) optimality, theoretical convergence of the proposed algorithm.
- The authors establish strong and weak consistency under DC-SBM, ensuring that the method asymptotically recovers the true cluster labels.
- The method is extensively tested on benchmark datasets (Cora, CiteSeer, PubMed), as well as larger real-world datasets. And the results highlight the method’s superiority over both GNN-free methods and GNN-based architectures.

---

> ### Author Response · Authors · 2025-03-01
> **Response by authors (part 1 of 3)**
>
> We appreciate the reviewer’s positive feedback and for recognizing the strengths of our paper, including the convergence and consistency guarantees alongside empirical evaluations and integration with deep learning architectures. Their recognition of these aspects is greatly encouraging. We also want to thank the reviewer for the detailed and thorough feedback, which has significantly contributed to improving our paper.
>
> ## Major Concerns
> ### 1.
> **The paper reads more like a Ph.D. thesis rather than a journal article. The attempt to cover too many aspects makes it difficult to read smoothly, and it is challenging to follow through to the end. A better approach might be to feature the core findings in this paper and consider splitting other aspects, such as convergence analysis and consistency analysis, into a sequel paper.**
>
> We appreciate the feedback and have made significant revisions to improve readability and streamline the presentation. To ensure a more focused narrative, we have merged and shortened the background and literature review sections, allowing the core contributions to stand out more clearly. Since our primary contribution lies in integrating modularity maximization with graph coarsening, we have refined the justification for our optimization formulation, ensuring that the consistency analysis on DC-SBMs supports this argument. While the convergence of the Block MM framework is novel in the context of graph clustering, it is not the focal point of our work. However, removing it entirely would weaken the completeness of our approach. Instead, we have moved the full proof of the theorem to Appendix G while providing a concise proof outline in the main text. Additionally, we have reorganized sections, rewritten portions of the second section for better coherence, and refined transitions to improve the overall flow. These revisions enhance readability while maintaining a strong, cohesive argument without fragmenting the contributions into separate papers.
>
>
> ### 2.
> **The proposed optimization framework is quite similar to Kumar's (2023, JMLR), except for the addition of the modularity term ( tr(C^TBC) ) in the objective function—Eq. (5). By incorporating this term, the problem is no longer convex, leading the authors to propose an optimization approach using MM. Therefore, the distinction from Featured Graph Coarsening (FGC) in Kumar’s work should be clearly highlighted. Additionally, comparative studies should be conducted more thoroughly, rather than treating FGC as merely one of the comparative methods in the current experimental setup.**
>
> We agree with the reviewer that our method is inspired by the FGC framework. However, there are significant differences between the two approaches. Our results show that FGC alone does not perform effectively for graph clustering. A key contribution of our work is the use of modularity maximization and demonstrating the consistency of our method. Additionally, integrating our objective as a loss function with GNNs and VGAEs to achieve state-of-the-art performance is another central contribution.
>
> We have also conducted ablation studies (Table 4 in Section 5.5) comparing the conductance and modularity of FGC with our method. The results clearly demonstrate that FGC is not well-suited for graph clustering.
>
>
> ### 3.
> **Particularly, the ablation study for each term in the loss function needs to be emphasized more clearly, rather than being placed in the Appendix as in the current paper. Additionally, the legends in Figure 7 are difficult to recognize. Please improve their readability and check the font size of other figures to ensure clarity.**
>
> We thank the reviewer for this suggestion. We have now plotted the figure showing evolution of loss terms again to improve readability. We have now placed it in the main paper under ablation studies Section 5.5 (Figure 3b).
>
> ### 4.
> **The related works presented in the Introduction and Section 2: Related Works are somewhat confusing. Narrow the scope, present them in a more integrated manner, and explicitly explain their relevance to the proposed method.**
>
> We have merged the background section with the related works to streamline the content. We refer the reviewer to Section 2 of the latest revision. The related works have been shortened, focusing on previous studies that are directly relevant to understanding our method. A more comprehensive literature review is available in the Appendix B.

---

> ### Author Response · Authors · 2025-03-01
> **Response by authors (part 2 of 3)**
>
> ### 5.
> **In the paragraph below Eq.(4), the coarsening objective ( f ) is described as preserving inter-connectivity, while in Eq.(5), ( h ) is stated to maintain inter-connectivity. This is confusing. Simply put, in the coarsened graph, ( f ) should be responsible for maintaining the smoothness of node features, while ( h ) should preserve inter-connectivity. However, why does increasing $( -\log \det (C^\top \Theta C + J) )$ lead to such effects? Please provide a more detailed explanation.**
>
> We have now corrected the paragraph below Equation 4 to the following:
> `f denotes the graph coarsening objective which compresses similar nodes together while preserving structural properties`
>
> The log-determinant term ensures that the coarsened graph is connected - i.e. preserving inter-cluster relations. Because it can be written as $-\sum_i\log\lambda_i$ where $\lambda_i$'s are the eigenvalues of the $\Theta_C$, this makes it so that $\Theta_C$ has minimal multiplicity of 0-eigenvalues (which tells us how many connected components there are, and we want just 1 big connected component).
>
> <!-- By minimizing this, we are ensuring that minimal number of $\lambda_i$'s are 0, since the number of connected components in a graph is equal to the multiplicity of 0 in it's laplacian eigenvalues. -->
>
> We have discussed this in Section 3.
>
> ### 6.
> **To show the (f(C)) in Eq.(6) is L-Lipschitz continuous gradient function', each term of which is proved as a L-Lipschitz continuous' in Appendix B. However, Lipschitz continuity does not necessarily means Lipschitz-Continuous Gradient of (f(C)). So the more explanation is needed.**
>
> We thank the reviewer for highlighting this mistake. We only require f(C) to be Lipschitz continuous, not a Lipschitz continuous gradient function. This was a typing error on our part, and we apologize for the oversight. The correction has been made in the updated revision.
>
> ### 7.
> **It is difficult to follow the transition to Eq(8). How is Eq.(8) formulated as a majorized problem of Eq.(5)? Since one of the main contributions of the proposed method is the majorization of the objective function, a more detailed explanation of the procedure should be provided.**
>
> We define the majorized function as g(C∣C_t), treating C_t as a constant. By simplifying g(C|C_t), we remove all constant terms and retain only the terms that depend on C. We have now added this derivation to the main paper Section 3 (Equations 7-10). We would be happy to answer any further questions and are open to making additional changes to improve readability if needed.
>
> $$
> \begin{align}
>     g(C|C^t) &= f(C^t) + \nabla f(C^t) \cdot (C - C^t) + \frac{L}{2} \text{tr}\big((C - C^t)^T (C - C^t)\big)\\\\
>     &=  f(C^t) + \nabla f(C^t) \cdot (C - C^t) + \frac{L}{2} \text{tr}(C^T C) - L \text{tr}(C^T C^t) + \frac{L}{2} \text{tr}((C^t)^T C^t)\nonumber\\\\
>     \text{Ignoring}& \text{ constant terms we get}\nonumber\\\\
>     &= \text{tr}(C^T \nabla f(C^t)) - L \text{tr}(C^T C^t) + \frac{L}{2} \text{tr}(C^T C)
> \end{align}
> $$
>
> Now, the majorised problem of Eqn. 6 becomes
> $$
> \begin{align}
>     \underset{C \in \mathcal{S}_C}{\min}\ \text{tr}(\frac{1}{2}C^TC - C^T\Big(C^t - \frac{1}{L}\nabla f(C^t)\Big)\Big)
> \end{align}
> $$
>
> ### 8.
> **In Appendix B, Equation (62) defines the optimal solution satisfying the KKT conditions as:  $C = \frac{(A)^+}{\sum_i ||[A^T]_i||_2}$. However, ( A ), as defined below Equation (55), is given by: $A = \left( C - \frac{1}{L} \nabla f(C) \right)^+$ which is a recursive formulation. How can this be solved?**
>
> We have now clarified these as Equations 47-55 in Appendix D.
> Q-MAGC is an iterative algorithm. Hence,
>
> $C^{(t+1)}_{\text{un}} =  \Big(C^{(t)} - \frac{1}{L}\nabla f(C^{(t)}) \Big)^{+} $
>
> is the unnormalized $C$ after $t^{th}$ majorizing iteration.
> $$
> C^{(t+1)} = \frac{ (C^{(t+1)}_{\text{un}})^{+} }{ \text{norm} }
> $$
>
> where, $ \text{norm} = \sum_i \bigg\|\bigg\| [{C^{(t+1)}_{\text{un}}}^T]_i \bigg\| \bigg\|_2 $
>
> So, $A$ has been renamed to $C^{(t+1)}_{un}$.
> > Note: In the paper, we have written it more concisely.
>
> ### 9.
> **Moving the Lagrangian equation (Eq. 55) from Appendix B to the main manuscript will enhance the readability of the proposed method.**
>
> We thank the reviewer for the suggestion. We have now added a proof sketch for the convergence analysis in the main paper in Section 3 and referenced the Lagrangian equation to aid readers in understanding the analysis. However, due to space constraints, we did not include the full equation in the main paper.
>
> ### 10.
> **The section on consistency analysis with DC-SBM takes up too much space. It would be better to shorten this part, similar to how convergence analysis and complexity analysis are presented.**
>
> We have now moved the proof to Appendix G and added a proof sketch in the main paper (after Theorem 2). We thank you for this suggestion.

---

> > ### Author Response · Authors · 2025-03-01
> > **Response by authors (part 3 of 3)**
> >
> > ### 11.
> > **Hyperparameter tuning and data preprocessing details are not sufficiently discussed, making replication and comparison more difficult.**
> >
> > We have added details about hyperparameter tuning and preprocessing in Appendix K  of the latest revision. We thank you for this suggestion.
> >
> > ### 12.
> > **Important figures and tables for key experiments are placed in the appendix. Switching the proofs to Appendix and moving the experimental setup and results to the main text would provide better accessibility for a wider range of readers.**
> >
> > We thank the reviewer for this suggestion. We have now moved the ablation of evolutions of loss terms with training to the main paper. Additionally, we have moved the consistency proof to Appendix G.
> >
> > ### 13.
> > **The authors acknowledge that the proposed approach relies heavily on modularity. In datasets where modularity is naturally low (e.g., the Airports dataset), the performance advantage over other methods diminishes. This raises concerns about the generalizability of the method to graphs with weakly modular structures.**
> >
> > Yes, the performance advantages are not prominent in low-modularity datasets. However, our method still remains competitive and manages to perform equally well. The importance of the other terms is highlighted in this case.
> > From Table 3, we see that our methods manage to reach within 1.5 on Barzil and surpass the SOTA by 1 point on Europe dataset.
> >
> > ### 14.
> > **Please revise the reference list to follow a standard format, either chronological or alphabetical, to improve clarity. Currently, it is difficult to locate specific citations within the list.**
> >
> > We are using the TMLR-provided citation format (`tmlr.sty`) with `natbib`. The papers are sorted according to the last name of the first authors.
> >
> > ### 15.
> > **Magnify the font size of markers and legends in Figure 2 for better readability. Additionally, Figure 2(a) is presented as a figure, but it is actually a table. The misuse of object types due to space constraints should be avoided. A similar issue is found in Figure 3, which should also be corrected. Ensure that figures and tables are appropriately categorized to maintain clarity and accuracy in presentation.**
> >
> > We have now categoried the figures and tables correctly. Additonally, we have enlarged the font size for the figures.
> >
> > ### 16.
> > **Align the appendix sections to match their order of appearance in the main manuscript.**
> >
> > We have reordered the appendices in the latest revision.
> >
> > ## Minors
> > 17. **On page 2, when Q-MAGC first appears, write out its full name and provide appropriate references.**\
> > We have now added the full form of Q-MAGC in the Introduction.
> >
> > 18. **On page 3, provide appropriate references for block maorization-minimization (Block MM) techniques.**\
> > References for Block MM have been added to the Introduction.
> >
> > 19. **Provide the full name for DC-SBM when it first appears.**\
> > The abstract and introduction already included the full form of DC-SBM when it was first mentioned. Based on the reviewer’s suggestion, we have now added the full form in the Section 3 as well.
> >
> > 20. **On page 5, when Q-MAGC first appears, write out its full name to ensure clarity for readers encountering it for the first time**\
> > We have now added the full name for Q-MAGC in the Introduction (Section 1).
> >
> > 21. **Eq. (5),(6) Typo: (C \in R^{p \times l}) ( l ) is not correct.**\
> > This was a typing mistake. We have replaced l with k.
> >
> > 22. **In Appendix A, Equation (29), add a `+` sign before the last term.**\
> > We have corrected this.
> >
> > 23. **In Appendix A, Equation (31), replace ( V ) with ( U ).**\
> > We have corrected this typing error.
> >
> > 24. **It is recommended that the Lipschitz constants ( L_1, L_2, L_3, L_4 ) be assigned in the order in which they appear in either equation (5) or (6).**\
> > We have now reordered the Lipschitz constants to match Equation 5.

---

> > > ### Author Response · Authors · 2025-03-01
> > > **Summary of changes in the latest revision and PDF diff**
> > >
> > > # Changes
> > > Based on the feedback received from the reviewers, we have made the following changes. We have uploaded a PDF with the changes colored (as the main submission PDF). A side-by-side diff PDF of the changes highlighted from the last version is available at https://anonymous.4open.science/api/repo/MAGC-8880/file/TMLR%20Final%20Diff.pdf
> > >
> > > ## Major Changes
> > > - Merged Background and Related Works sections to make it concise and improve readability.
> > > - Moved  the detailed literature survey to Appendix B.
> > > - Explain why log-det term helps with inter-connectivity.
> > > - Added small proof for deriving the majorized function (Equations 7-10).
> > > - Added a proof sketch for the convergence analysis.
> > > - Added reasons for choosing DC-SBM for the consistency analysis in Section 3.
> > > - Moved the consistency proof to Appendix G and added a proof sketch in the main paper.
> > > - Clarified the importance of the consistency analysis in the Introduction and Section 3.
> > > - Added a datasets table in the main paper (Table 1).
> > > - Added hyperparameter tuning details in Appendix K.1
> > > - Moved figure showing evolution of various loss terms with training from the appendix to the main paper (Figure 3b).
> > > - Added a hyperparameter sensitivity ablation study in Appendix K.2
> > > - Added a discussion on the visualization of latent spaces.
> > > - Added a discussion on the performance of Q-GCN/VGAE/GMM-VGAE.
> > > - Reordered the appendices based on first occurence.
> > >
> > > ## Minor Changes
> > > - Improved the captions of the figures and increased sizes.
> > > - Clarified meaning of Q-MAGC in the Introduction.
> > > - Fixed minor grammatical and typing errors in the paper.

---

### Decision · Action_Editor_4KQp · 2025-04-29

**Recommendation:** Accept as is

**Comment:**

First of all, I would like to thank the authors for their patience during the unusually long review process.
For various reasons, it took a long time to find reviewers, as many invitees either did not respond or declined the invitation.
Finally, I am pleased to provide a positive feedback to you.

**Audience:**

As pointed out in the abstract, graph clustering is an important unsupervised learning technique
for partitioning graphs with attributes and detecting communities.  The provably convergent and time-efficient algorithm proposed in this paper
may attract interests from researchers and practitioners workin on graph-structured data.

**Claims And Evidence:**

This paper addresses graph clustering and presents a method which integrates graph coarsening and modularity maximization. All of reviewers feel that the paper provides theoretical soundness and convergence guarantee of the proposed optimization framework, which well justifies the evidence of why the method works. In addition, empirical results demonstrate the validity of the method, highlighting its superiority over both GNN-free methods and GNN-based architectures.

---

> ### Author Response · Authors · 2025-06-06
>
> We are grateful to our action editor for their unwavering support throughout the review process. We also extend our sincere thanks to all the reviewers for their thorough and encouraging feedback. We appreciate the level of detail that reviewers went into in their evaluation of our paper.
>
> Thank you!\
> ~ Authors